# GEOMETRY-AWARE POLICY IMITATION

**Yiming Li**[*†]    **Nael Darwiche**[†]    **Amirreza Razmjoo**[*†]    **Sichao Liu**[†]

**Yilun Du**[‡]    **Auke Ijspeert**[†]    **Sylvain Calinon**[*†]

[*] Idiap Research Institute    [†] EPFL    [‡] Harvard University

## ABSTRACT

We propose a Geometry-aware Policy Imitation (GPI) approach that rethinks imitation learning by treating demonstrations as geometric curves rather than collections of state–action samples. From these curves, GPI derives distance fields that give rise to two complementary control primitives: a progression flow that advances along expert trajectories and an attraction flow that corrects deviations. Their combination defines a controllable, non-parametric vector field that directly guides robot behavior. This formulation decouples metric learning from policy synthesis, enabling modular adaptation across low-dimensional robot states and high-dimensional perceptual inputs. GPI naturally supports multimodality by preserving distinct demonstrations as separate models and allows efficient composition of new demonstrations through simple additions to the distance field. We evaluate GPI in simulation and on real robots across diverse tasks. Experiments show that GPI achieves higher success rates than diffusion-based policies while running 20× faster, requiring less memory, and remaining robust to perturbations. These results establish GPI as an efficient, interpretable, and scalable alternative to generative approaches for robotic imitation learning. Project page: https://yimingli1998.github.io/projects/GPI/

## 1 INTRODUCTION

Robots are increasingly expected to perform complex tasks in unstructured environments, ranging from dexterous manipulation to interactive collaboration. *Imitation learning* offers a promising path toward this goal, as it enables robots to acquire policies directly from expert demonstrations without relying on explicit dynamics models or simulation. Existing imitation approaches can be grouped into three families. *Explicit policies* treat imitation as supervised regression from states to actions (Calinon et al., 2007). They are fast at inference but struggle with multimodality and generalization. *Implicit policies* learn energy functions over state–action pairs (Florence et al., 2022), but are hard to train and slow to optimize at deployment. *Generative policies*, such as diffusion or flow-matching models (Chi et al., 2023; Lipman et al., 2023), excel at modeling multimodality but remain computationally heavy and brittle under distribution shifts. Despite their differences, all three approaches compress demonstrations into parametric models that must be retrained to incorporate new data and that often discard the geometric structure underlying expert behavior.

We argue that imitation learning can be made more direct, interpretable, and efficient by adopting a *geometric approach*. At its core, imitation means: (i) following the expert's direction of motion, while (ii) approaching expert states as closely as possible. Viewed this way, a demonstration is not just a collection of samples but a *geometric curve* in state space, annotated with tangents that indicate expert actions. This perspective motivates our approach, **Geometry-Aware Policy Imitation (GPI)**. GPI represents demonstrations as *distance fields* that can be projected onto the robot's actuated subspace, where control is applied. From these fields naturally emerge two complementary primitives: a *progression flow* that advances along expert trajectories, and an *attraction flow* that pulls current states toward them. Superimposing these flows defines a controllable vector field that drives imitation (Li & Calinon, 2025). This approach provides an approximation that reduces deviation while advancing along expert behaviors (Figure 1). In addition, the policy is guided by a distance field composition that retrieves flow fields from the most similar demonstrations, promoting coherent behavior and enabling robustness even under unknown dynamics.

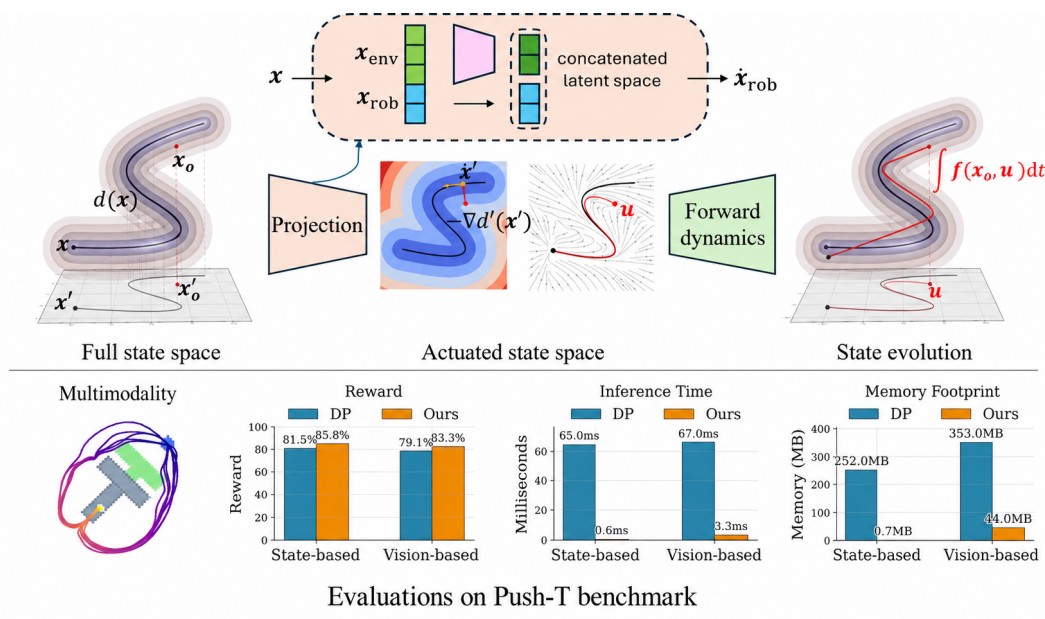

Figure 1: **Overview of Geometry-Aware Policy Imitation (GPI).** GPI treats demonstrations as geometric curves that induce distance fields in the full state space. (**Top**) The state space is projected onto the robot's actuated subspace, where control is applied. The projected distance field gives rise to two complementary flows: an *attraction flow* from the negative gradient (red arrow) and a *progression flow* from trajectory tangents (yellow arrow). Together, they define a dynamical system that reduces the distance to demonstrations and advances along them, thus imitating expert behavior. The resulting action $u$ is executed through the system's dynamics, yielding state evolution $\int f(x, u)\, dt$ in the full state space. Multiple demonstrations can be composed naturally via Boolean operations on distance fields. Despite unknown system dynamics, the resulting trajectory aligns closely with the most similar demonstration as determined by the distance metric. (**Bottom**) On the PushT benchmark, GPI achieves multimodal imitation with a higher reward, runs 20–100× faster than diffusion policies (DDIM with 10 steps), and requires substantially less memory.

A key strength of GPI is its *decoupling* of imitation into two modular components: (i) **metric learning**, which defines how states are represented and compared; and (ii) **behavior synthesis**, which constructs policies directly from distance and flow fields. This separation offers substantial flexibility: low-dimensional states can use Euclidean or geodesic distances, while high-dimensional observations can rely on latent embeddings from pretrained or task-specific encoders. Policy synthesis itself is non-parametric and lightweight, enabling efficient composition of demonstrations without retraining and supporting multimodality by preserving distinct trajectories as separate flows (Pari et al., 2022). Moreover, because GPI only requires a state representation that supports distance computation, rather than directly fitting a full policy function, the learning problem is considerably simpler than in generative models. Lightweight encoders are typically sufficient, which reduces training complexity and enables fast inference at deployment.

We evaluate GPI extensively in both simulation and on real robots. In simulation, we benchmark across diverse domains—including planar pushing, 6-DoF manipulation, and dexterous hand control—with state spaces ranging from low-dimensional control vectors to raw vision inputs. For visual observations, we study multiple feature representations, from pretrained encoders to self-supervised embeddings. On real hardware, we demonstrate GPI on both a Franka arm and the Aloha bimanual system, showing that it scales robustly beyond controlled environments.

In summary, our contributions are:

i) **Geometry-Aware Policy Imitation (GPI)**, which represents demonstrations as geometric curves that induce composable distance fields, providing a unified representation for both metric reasoning and action synthesis;

ii) **A simple and modular formulation**, where state representation relies only on a suitable distance metric and action synthesis is realized through compositions of control primitives. Both components are lightweight, flexible, and grounded in well-studied principles;

iii) **Extensive validation** in simulation and on real robots, showing that GPI achieves higher performance and enables efficient policy imitation—over $20\times$ faster than state-of-the-art diffusion policies—while remaining interpretable and multimodal.

## 2 GEOMETRY-AWARE POLICY IMITATION

GPI constructs policies directly from demonstrations by representing them as geometric curves in state space. Each demonstration induces a distance field that encodes state similarity and gives rise to two complementary control primitives: (i) a *progression flow* that advances along demonstrated motions, and (ii) an *attraction flow* that corrects deviations by pulling states toward the trajectory. Their superposition defines a dynamical system that imitates expert behavior. Local policies derived from individual demonstrations are then composed via distance-based weighting, producing a coherent global policy that is efficient, interpretable, and robust to perturbations. Figure 1-*top* illustrates these components schematically.

### 2.1 METHOD

We are given $N$ expert demonstrations $\mathcal{D} = \{\Gamma^{(i)}\}_{i=1}^N$, where each $\Gamma^{(i)}$ is a trajectory consisting of a sequence of states and actions

$$\Gamma^{(i)} = \{(\boldsymbol{x}_t^{(i)}, \boldsymbol{u}_t^{(i)})\}_{t=0}^{T_i}, \tag{1}$$

with states $\boldsymbol{x}_t^{(i)} \in \mathcal{X}$, actions $\boldsymbol{u}_t^{(i)} \in \mathcal{U}$, and horizon $T_i$.

**State and actuated subspace.** A state $\boldsymbol{x}$ may include both environment variables (e.g., object poses, images) that are unactuated, and robot variables that are directly actuated by control inputs. We denote by $\boldsymbol{x}' = P(\boldsymbol{x})$ the projection of $\boldsymbol{x}$ onto the actuated subspace $\mathcal{X}' \subseteq \mathcal{X}$, where $P : \mathcal{X} \to \mathcal{X}'$ is the projection operator. Each trajectory $\Gamma^{(i)}$ can then be viewed as a geometric curve in state space, which induces a *distance field* $d(\boldsymbol{x}_o \mid \Gamma^{(i)})$ measuring the proximity between a query state $\boldsymbol{x}_o$ and the demonstration.

**Action space.** We assume *velocity control* in the actuated subspace, i.e., $\boldsymbol{u}_t = \dot{\boldsymbol{x}}_t'$. Each demonstration $\Gamma^{(i)}$ then defines a curve $\boldsymbol{x}_t^{(i)}$ whose actions $\boldsymbol{u}_t^{(i)}$ are the tangent directions in $\mathcal{X}'$. Velocity control is used here for clarity, but it is not a prerequisite: the formulation extends naturally to accelerations or torques, which can be executed through the robot's kinematics or dynamics models.

**Policy as flow field in actuated space.** From the distance field $d(\boldsymbol{x}_o \mid \Gamma^{(i)})$ induced by a demonstration $\Gamma^{(i)}$, we derive two complementary flows in the actuated subspace: *Progression flow*, given by the demonstrated tangent action $\boldsymbol{u}_{\kappa(\boldsymbol{x}_o)}^{(i)} = \dot{\boldsymbol{x}}_{\kappa(\boldsymbol{x}_o)}'^{(i)}$, which advances along the expert trajectory; and *Attraction flow*, obtained from the partial derivative of the distance field with respect to actuated coordinates, $-\nabla_{\boldsymbol{x}_o'} d(\boldsymbol{x}_o \mid \Gamma^{(i)})$, which corrects deviations by pulling states back toward demonstrations. Their superposition defines a policy in the actuated subspace:

$$\pi_i(\boldsymbol{x}_o) = \lambda_1(\boldsymbol{x}_o)\, \boldsymbol{u}_{\kappa(\boldsymbol{x}_o)}^{(i)} - \lambda_2(\boldsymbol{x}_o)\, \nabla_{\boldsymbol{x}_o'} d(\boldsymbol{x}_o \mid \Gamma^{(i)}), \tag{2}$$

where $\kappa(\boldsymbol{x}_o) = \arg\min_t d(\boldsymbol{x}_o, \boldsymbol{x}_t^{(i)})$ denotes the nearest demonstrated state, and $\lambda_1, \lambda_2 \geq 0$ are weights—either constant or distance-dependent chosen so that attraction dominates far from demonstrations, while progression dominates near them. This policy has been shown to yield a stable first-order dynamical system that asymptotically converges to the demonstrated trajectory if the state and action variables are continuous (Li & Calinon, 2025)[1]. This can be achieved by representing a discrete trajectory with continuous functions such as splines. Thus, the robot's behavior remains robust, predictable, and safe even under environmental changes or perturbations.

**Composition across demonstrations.** To obtain a global policy, we compose local flow-based policies across multiple demonstrations. Given the $K$ nearest demonstrations, the global policy is

$$\pi(\boldsymbol{x}_o) = \sum_{i=1}^K w_i(\boldsymbol{x}_o)\, \pi_i(\boldsymbol{x}_o), \qquad w_i(\boldsymbol{x}_o) = \frac{\exp\big(-\beta\, d(\boldsymbol{x}_o \mid \Gamma^{(i)})\big)}{\sum_{j=1}^K \exp\big(-\beta\, d(\boldsymbol{x}_o \mid \Gamma^{(j)})\big)}, \tag{3}$$

---

[1]See Appendix A for the proof.

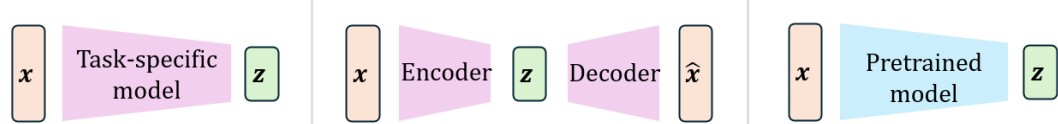

Figure 2: **Typical ways to obtain latent embedding $z$ from raw inputs $x$.** (i) train a task-specific lightweight model to capture task-relevant features; (ii) use a VAE to learn task-agnostic features; or (iii) apply a pretrained model to obtain features without additional training.

where $\pi_i(\boldsymbol{x}_o)$ is the local policy induced by demonstration $\Gamma^{(i)}$, $d(\boldsymbol{x}_o \mid \Gamma^{(i)})$ is the distance from query state $\boldsymbol{x}_o$ to the trajectory $\Gamma^{(i)}$, and $\beta > 0$ is a temperature parameter controlling the sharpness of selection. This distance-based composition ensures that flows are retrieved from the most relevant demonstrations, yielding coherent behavior even under unknown dynamics. A detailed description of GPI is provided in Algorithm 1 (Appendix B).

## 2.2 CHOICE OF DISTANCE METRIC

A central design choice in GPI is the distance metric $d(\boldsymbol{x}_o \mid \Gamma^{(i)})$, which measures the similarity between a query state and a demonstration. The state naturally consists of two complementary parts: the robot-actuated variables (e.g., joint angles, end-effector pose) and the environment-related variables (e.g., object poses, images). Accordingly, the distance metric can be decomposed into a robot feature $d_{\text{rob}}$ and an environment feature $d_{\text{env}}$, where the former also shapes the attraction flow in actuated space and the latter only influences demonstration selection and weighting.

**Robot distance $d_{\text{rob}}$.** For joint or end-effector positions $\boldsymbol{x} \in \mathbb{R}^n$, Euclidean distance is standard:

$$d_{\text{Euc}}(\boldsymbol{x}_1, \boldsymbol{x}_2) = \|\boldsymbol{x}_1 - \boldsymbol{x}_2\|_2. \tag{4}$$

For end-effector orientations represented as quaternions, geodesic distances on $S^3$ respect rotational geometry:

$$d_{\text{quat}}(\boldsymbol{x}_1, \boldsymbol{x}_2) = 2 \arccos(|\langle \boldsymbol{x}_1, \boldsymbol{x}_2 \rangle|). \tag{5}$$

These two cases cover the most common representations in joint space and task space for robotics.

**Environment distance $d_{\text{env}}$.** This compares task-relevant but indirectly controllable variables, such as object poses or scene images. For low-dimensional object poses, $d_{\text{env}}$ can be computed with Euclidean or geodesic distances, reusing the formulations above. For high-dimensional observations, it is common to define $d_{\text{env}}$ in a latent space. Let $\boldsymbol{z} = \Psi(\boldsymbol{x})$ denote the latent embedding of $\boldsymbol{x}$. Then

$$d_{\text{env}}(\boldsymbol{x}_1, \boldsymbol{x}_2) = d_{\text{env}}(\boldsymbol{z}_1, \boldsymbol{z}_2), \tag{6}$$

where $\boldsymbol{z}_1 = \Psi(\boldsymbol{x}_1)$ and $\boldsymbol{z}_2 = \Psi(\boldsymbol{x}_2)$ are latent embeddings produced by a parametric model $\Psi$ that maps raw observations to a latent space, and $d(\cdot, \cdot)$ denotes a suitable distance (e.g., Euclidean or cosine). This formulation supports multiple sources of embeddings: (i) task-specific models, where $\boldsymbol{z}$ could encode predicted object poses or desired robot actions learned via supervision; (ii) latent variables from variational autoencoders (VAEs) trained with self-supervised objectives (Kingma & Welling, 2013); and (iii) pretrained vision or multimodal encoders such as SAM (Kirillov et al., 2023), DINO (Siméoni et al., 2025), and CLIP (Radford et al., 2021), see Figure 2 for an overview. Classical dimensionality-reduction methods, such as principal component analysis (PCA), can also be used to obtain a compact latent feature (Hotelling, 1933).

While both $d_{\text{rob}}$ and $d_{\text{env}}$ contribute to the overall distance metric, their roles differ: $d_{\text{env}}$ influences only the similarity ranking across demonstrations, whereas $d_{\text{rob}}$ additionally shapes the attraction flow in the actuated subspace. This decomposition makes explicit how environmental features guide demonstration selection, while robot features govern the actual corrective control.

## 2.3 A 2D EXAMPLE

To illustrate GPI, we consider a simplified 2D setting where the state consists only of actuated variables $\boldsymbol{x}'$. This abstraction is common in kinematic planning tasks, where environment dynamics are ignored. In this case, the distance field reduces to the robot-related term, $d(\boldsymbol{x}_o) = d_{\text{rob}}(\boldsymbol{x}'_o)$, so that state evolution and policy flows are fully contained in the same space. While prior work typically trains diffusion or flow-matching models for policy generation in this setting (Jiang et al.,

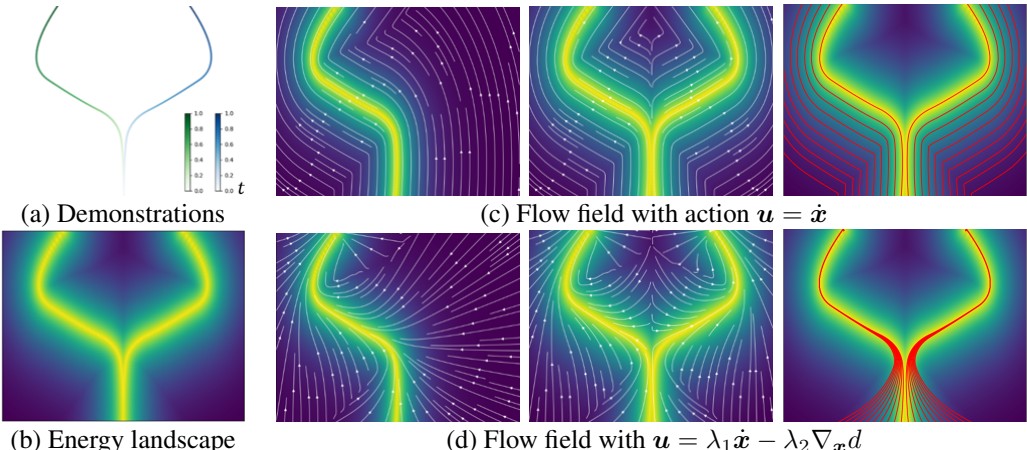

(a) Demonstrations

(c) Flow field with action $\boldsymbol{u} = \dot{\boldsymbol{x}}$

(b) Energy landscape

(d) Flow field with $\boldsymbol{u} = \lambda_1\dot{\boldsymbol{x}} - \lambda_2\nabla_{\boldsymbol{x}}d$

Figure 3: **From demonstrations to policy flows.** (a) Demonstrations. (b) Energy from composed distances. (c) Progression-only flow $\boldsymbol{u} = \dot{\boldsymbol{x}}$ may drift off the demonstrations. (d) Adding attraction $\boldsymbol{u} = \lambda_1\dot{\boldsymbol{x}} - \lambda_2\nabla_{\boldsymbol{x}}d$ pulls states toward the demonstrations and along them, ensuring convergence.

2025), GPI instead addresses the problem in a fully non-parametric manner, relying directly on the distance and flow fields.

Figure 3(a) shows two demonstrations forming a Y-shaped pattern: $\Gamma^{(1)}$ (green) and $\Gamma^{(2)}$ (blue) overlap initially and then diverge into separate branches. Temporal progression is indicated by transparency from $t = 0$ to $t = 1$. Each demonstration induces a Euclidean distance field whose valleys align with its trajectory; composing them yields a global distance field (Figure 3b) visualized as an energy landscape with dense corridors along the demos and a natural decision boundary at the bifurcation. Figures 3(c,d) show the resulting flow fields: each row includes the single-demo flow (left) and the composed flow (both demos), with rollout trajectories overlaid on the energy landscape (right). Panel (c) depicts the progression flow, which follows the local tangent of the nearest demonstration; Panel (d) augments this with an attraction term that pulls states toward the trajectories, ensuring stable convergence. The rollout trajectories (red) show the integrated trajectories in two cases. From this perspective, diffusion policies perform well because their denoising steps implicitly induce an attraction flow toward demonstrations rather than relying solely on progression.

By representing demonstrations as distance and flow fields, policy imitation shifts from fitting a parametric model to geometric reasoning grounded in similarity, curvature, and composition, yielding several benefits: **Efficiency**—new demonstrations enrich the distance field by adding basins of attraction without retraining, and inference reduces to distance evaluations plus weighted averaging of expert actions, making it lightweight and parallelizable; **Flexibility**—decoupling similarity measurement from action synthesis keeps the framework modular, allowing task-specific distance metrics and flow compositions; **Multimodality**—each demonstration defines its own distance and flow field, preserving distinct behaviors so the policy branches smoothly toward the nearest demonstrated mode instead of averaging conflicting actions; **Interpretability**—the distance metric reveals which demonstrations influence the current action, while actions remain a linear superposition of demonstrated behaviors and corrective flows, ensuring safe, bounded outputs.

## 3 EXPERIMENTAL RESULTS

### 3.1 SIMULATION EXPERIMENTS

We first evaluate GPI on the PushT benchmark, a widely adopted task in which a robot must push a T-shaped object into a target configuration (Chi et al., 2023). This environment is particularly suitable for evaluation: it has well-established baselines for comparison, requires handling inherently multimodal pushing strategies, and involves contact-rich dynamics that cannot be solved by simple kinematic planning.

For state-based inputs, demonstrations consist of the agent position, the object position, and the object orientation. Distances are computed as a weighted combination of these components. The actuated subspace corresponds to the agent position, with its first-order derivative (velocity) serving

as the action. Note that the original environment specifies actions in position control, which we adapt to velocity control for consistency with our flow-based formulation. Control policies are synthesized from the flow fields induced in the actuated subspace by corresponding demonstrations, and then executed in the environment with unknown interaction dynamics. For vision-based inputs, the state comprises the agent pose and an RGB image. Distances are computed jointly over the agent pose and an image embedding. To align with the state-based formulation, we train a lightweight task-specific model to produce the image embedding as the predicted object pose.

Table 1: Performance comparison on Push-T (state-based vs. vision-based).

| Method | Push-T (state-based) | | | Push-T (vision-based) | | |
| | (Avg./Max.) score (%) | Training / Inference Time | Memory | (Avg./Max.) score (%) | Training / Inference Time | Memory |
|---|---|---|---|---|---|---|
| DDPM | 82.3 / 86.3 | 1.0 h / 641 ms | 252 MB | 80.9 / 85.5 | 2.5 h / 647 ms | 353 MB |
| DDIM | 81.5 / 85.1 | 1.0 h / 65 ms | 252 MB | 79.1 / 83.1 | 2.5 h / 67 ms | 353 MB |
| FMP | 77.6 / 80.2 | 1.0 h / 58 ms | 251 MB | 75.1 / 79.3 | 2.5 h / 60 ms | 352 MB |
| SFP | 83.1 / 87.8 | 0.8 h / 51 ms | 240 MB | 77.5 / 81.2 | 2.0 h / 55 ms | 341 MB |
| GPI (Ours) | **85.8 / 89.0** | **0 h / 0.6 ms** | **0.7 MB** | **83.3 / 86.9** | **0.3 h / 3.3 ms** | **44 MB** |

Experiments were conducted on an NVIDIA RTX 3090 GPU. Further details appear in Appendices C.1 and C.2. We report performance using three complementary metrics: (i) *Average / maximum reward*, evaluated over multiple random seeds and environment variations, following the same protocol as the baselines; (ii) *time*, including training time and per-step inference time; and (iii) *memory footprint*, including memory cost for model parameters and stored demonstrations. We compare GPI against Diffusion Policy (DP) (Chi et al., 2023), Flow Matching Policy (FMP) (Zhang & Gienger, 2024), and Streaming Flow Policy (SFP) (Jiang et al., 2025). All baselines are implemented using their official public codebases. For DP, we evaluate both a 100-step Denoising Diffusion Probabilistic Model (DDPM) (Ho et al., 2020) and a 10-step Denoising Diffusion Implicit Model (DDIM) (Song et al., 2021). Following standard practice, DP and FMP predict an $H$-step action sequence ($H = 8$), whereas SFP and GPI naturally support reactive planning and operate with a one-step horizon ($H = 1$). Results are summarized in Table 1. GPI achieves higher success rates across all tasks than these baselines while being substantially more computationally efficient.

In the state-based setting, inference involves only low-dimensional, non-parametric distance evaluations and flow field composition, resulting in a latency of $0.6$ ms—nearly $100\times$ faster than diffusion- or flow matching–based baselines. Although GPI requires storing all demonstrations for distance measurement, the overall memory footprint remains lower than that of training large neural policies[2] Moreover, the underlying computations are lightweight and naturally parallelizable, further contributing to its efficiency. For vision-based inputs, we employ a ResNet-18 encoder trained solely for feature extraction rather than precise action prediction, which simplifies training and improves efficiency. As a result, training completes in only $0.3$ hours and inference runs at $3.3$ ms per step. Memory requirements are also reduced, since we store only the lightweight encoder and latent embeddings of demonstrations rather than raw images or large policy networks. Additionally, this modular structure allows the visual encoder to be reused across different tasks.

We further conduct a series of ablations to highlight the distinctive properties of GPI:

**Robustness.** We evaluate GPI's robustness along three complementary dimensions.

*Planning horizon:* GPI is reactive by default ($H = 1$), but it can also be extended to a receding-horizon scheme by updating the distance every $H$ steps. As shown in Figure 4, performance remains stable for horizons up to 16, showing GPI can operate either as a purely reactive controller (robust to external disturbances) or as a receding-horizon planner (with improved temporal consistency).

*Number of neighbors:* In action composition, we compare $K = 1, 3, 5, 10$. As shown in Fig-

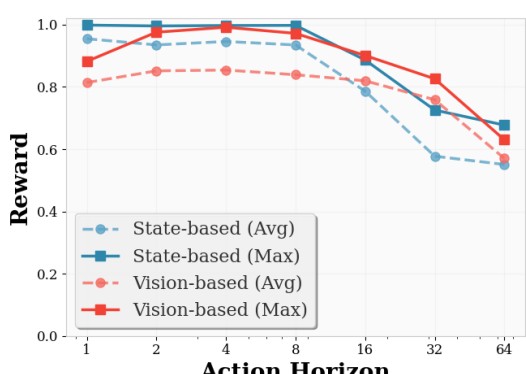

Figure 4: **Robustness to action horizons.**

---

[2]See Appendix D.1 for a detailed explanation.

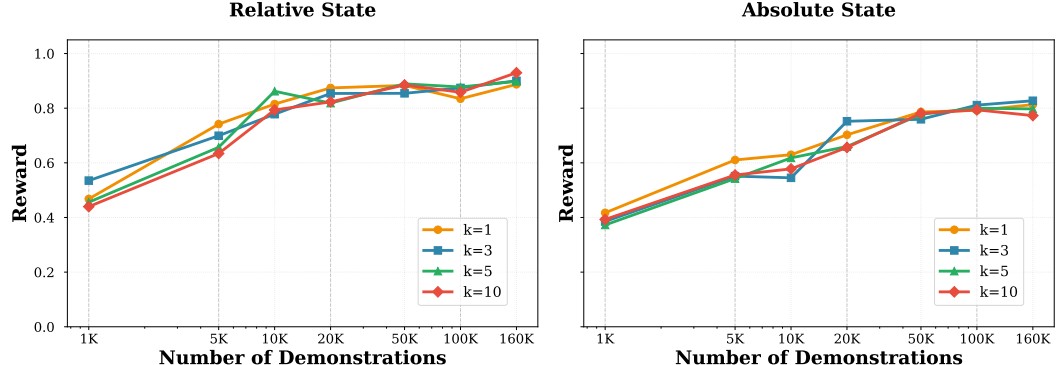

Figure 5: **Robustness of GPI with respect to demonstrations, $K$ (neighbors), and state representations.**

ure 5, the curves are nearly overlapping in both relative and absolute state settings, confirming that performance is largely insensitive to the choice of $K$. In addition, we also ablate the softmax temperature $\beta$ used in (3) for composing demonstrations, and the results are reported in Appendix D.4. This highlights the reliability of GPI's local composition mechanism.

*State representation:* We compare object-centric (relative) and global (absolute) state formulations (Figure 5). Both achieve strong performance, but relative states consistently yield slightly higher scores, especially in data-scarce regimes. This suggests that GPI is robust to representation choices, with relative states offering an advantage when demonstrations are limited.

*Distance metrics.* We evaluate how the choice of distance metric and the relative weighting between robot and environment components affect performance; detailed ablations and results are reported in Appendix D.3.

**Scalability with data sizes.** A distinctive advantage of GPI is that, being non-parametric and training-free in the state-based setting, it enables direct study of how performance scales with the number of demonstrations, without the need for retraining. To this end, we augment the dataset with up to 160K samples regenerated from the original diffusion policy work and evaluate how performance evolves as the demonstration set grows. This setting is particularly suitable for GPI, since demonstration density directly influences both the distance query and the selection of actions in the composed policy. As shown in Figure 5, success rates increase consistently as the dataset expands from 1K to 20K demonstrations, after which performance begins to saturate. This trend reveals two key insights: (i) larger demonstration sets provide denser coverage of the state space, thereby reducing approximation errors introduced by the chosen distance metric, and (ii) our approach can serve as a practical diagnostic tool—indicating how many demonstrations are sufficient to achieve reliable policy performance before training parametric models. The method also accommodates incremental incorporation of new demonstrations, without the need for full retraining. For a detailed analysis of latency and memory scalability as the number of demonstrations and the latent dimension increase, please refer to Appendix D.2.

**Stochasticity and multimodality.** To induce stochasticity and multimodality, we inject Gaussian noise $\mathcal{N}(0, \sigma^2)$ into the query state in the actuated space (corresponding to the agent's position). This perturbation alters the effective distance fields used in composition, thereby modifying the synthesized flow field and inducing multimodal behavior. In Figure 6, we compare the average score achieved under different noise levels. To quantify diversity, we measure the average distance among trajectories generated with different random perturbations sampled from the same noise distribution. The results show that larger noise values increase trajectory diversity but degrade perfor-

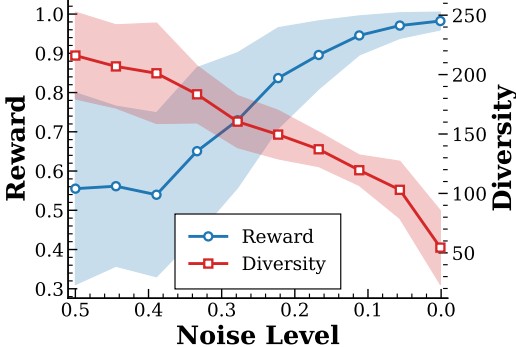

Figure 6: **Noise-level ablations for score and diversity.**

mance, whereas smaller noise levels yield more deterministic behavior. Importantly, GPI exhibits multimodal behavior even under low noise (e.g., $\sigma = 0.2$), as illustrated in Figure 1 (bottom left). Beyond Gaussian perturbations, stochasticity can also be enhanced by randomly subsampling the set of demonstrations at each inference time. We found that this strategy can improve performance in practice, for instance, by helping the robot escape from regions where it would otherwise become stuck.

**Natural composition of control primitives.** We interpret progression and attraction as two basic control primitives that can be naturally combined within the flow field. By varying their relative weights $(\lambda_1, \lambda_2)$, we interpolate between velocity-like (progression-driven) and position-like (attraction-driven) control. As shown in Figure 7, GPI maintains consistently high scores across a wide range of weightings, demonstrating flexibility in composing these primitives at test time rather than relying solely on fixed neural network outputs. In this view, progression promotes forward motion and task advancement, while attraction provides goal alignment and stability.

**Generalization across tasks.** We evaluate GPI on RoboMimic (Lift, Can, Square) (Mandlekar et al., 2021) and Adroit (Door, Pen, Hammer, Relocate) benchmarks (Rajeswaran et al., 2018), spanning state spaces of 9–46 dimensions and action spaces of 7–30. GPI consistently matches or exceeds the performance of Diffusion Policy without requiring any parametric training (Table 2), demonstrating robust generalization across diverse domains. The snapshots of those tasks are shown in Figures 11 and 12 (in Appendix D.5) respectively. Additionally, we test GPI on 2D Maze task (Chen et al., 2025; Janner et al., 2022) and visualization results in shown in Figure 13 in Appendix D.6.

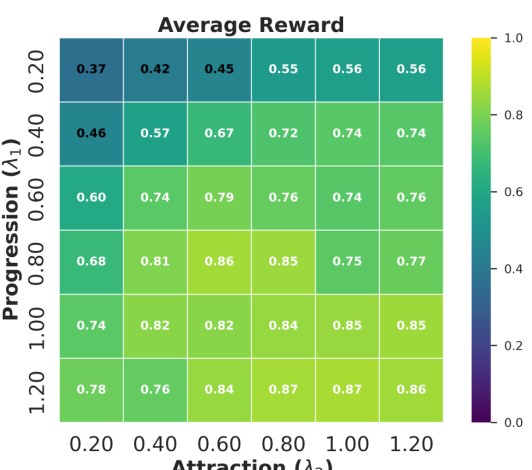

Figure 7: **Ablations on two control primitives.**

Table 2: Task description and performance on Robomimic and Adroit Hand benchmarks.

| | Task / Method | Robomimic | | | Adroit Hand | | | |
| | | Lift | Can | Square | Door | Pen | Hammer | Relocate |
|---|---|---|---|---|---|---|---|---|
| Description | State Dim | 9 | 16 | 16 | 39 | 45 | 46 | 39 |
| | Action Dim | 7 | 7 | 7 | 28 | 24 | 26 | 30 |
| | Demonstrations | 300 | 300 | 300 | 5000 | 5000 | 5000 | 5000 |
| Results | DP | **1.00** | 0.94 | **0.87** | **1.00** | 0.89 | 0.83 | **0.91** |
| | Ours | **1.00** | **0.96** | 0.82 | **1.00** | **0.95** | **0.88** | **0.91** |

**Generalization across visual representations.** As discussed in Section 2.1, GPI naturally accommodates multiple choices of latent embeddings, including task-specific encoders, VAEs, and pretrained models. We evaluate four variants on PUSHT: (i) a ResNet feature (He et al., 2016) pretrained within the Diffusion Policy implementation, with PCA applied for dimensionality reduction; (ii) an unsupervised variational autoencoder (VAE) trained solely on RGB images, serving as a task-agnostic feature extractor; and (iii) a pretrained Segment Anything (SAM) model (Kirillov et al., 2023) followed by a pose-estimation module whose predicted object pose serves as the embedding. Implementation details are provided in Appendices C.3 (ResNet+PCA), C.4 (VAE) and C.5 (SAM). Additionally, we compare our VAE-based encoder to a self-supervised Bootstrap Your Own Latent (BYOL) encoder (Grill et al., 2020) for feature extraction, which is also used in VINN (Pari et al., 2022) for non-parametric policy synthesis.

Results in Table 3 show that GPI with the same ResNet features followed by PCA achieves performance comparable to Diffusion Policy, which uses the same ResNet features with a diffusion head. Interestingly, a lightweight VAE encoder trained only for reconstruction also yields

Table 3: Performance of various visual representations on the pushT task.

| Feature Extractor | Avg. Score |
|---|---|
| Diffusion Policy | 85% |
| Task-specific Head | 87% |
| ResNet+PCA | 84% |
| VAE | **88%** |
| Pretrained SAM | 41% |
| BYOL feature | 67% |

strong performance. With the KL regularizer encouraging latents to stay near the prior $\mathcal{N}(0, I)$, it produces a smooth latent space in which linear interpolations tend to remain on-manifold. This VAE trains in $\sim 0.3$ hours and runs at $\sim 4$ ms per inference—comparable to our task-specific visual head (Table 1). In contrast, a self-supervised BYOL feature performs worse than the VAE on PUSHT. A plausible explanation is that the VAE's reconstruction objective encourages latent codes to retain and smoothly parameterize scene geometry, which is particularly well matched to GPI's distance fields and flows, whereas BYOL emphasizes invariance to augmentations and may discard some of this geometric information. Finally, even an off-the-shelf pretrained SAM model within the GPI framework achieves a 41% average score without any fine-tuning. This variant underperforms our other encoders, likely due to sensitivity to segmentation quality and the downstream pose estimation module; we expect that task-specific fine-tuning would improve its performance.

## 3.2 ROBOT EXPERIMENTS

To further evaluate GPI, we conduct robot experiments on two challenging tasks:

**(i) Box flip.** The robot must flip a box by exploiting contacts among the end-effector, the box, and an aluminum crossbeam, which is challenging due to unknown, highly nonlinear dynamics. We collect 121 demonstrations on an ALOHA platform (Aldaco et al., 2024). The dataset contains over 50,000 RGB images and action pairs. A lightweight neural network takes a raw RGB image as input and predicts an action; this predicted action serves as the image embedding. Distances are

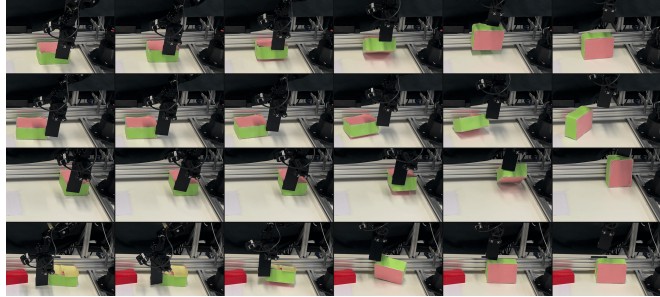

Figure 8: **Real-robot flipping task.** GPI successfully completes the task via multimodal behavior (Top 3 rows) and demonstrates robustness to visual disturbances (Bottom).

computed jointly over the robot joint configuration and the action embedding to construct the distance field, from which the flow field is derived for the robot's execution. We observe an inference time of approximately 7 ms and a memory footprint of 140 MB, comprising 139 MB for the feature-extraction model and 1 MB for storing latent features. In 50 flip trials, 39 are successful, corresponding to a 78% success rate. A trial is counted as successful if the robot flips the box to the target orientation within 500 control steps at 50 Hz. During these experiments, we also introduce occlusions and external disturbances; GPI still reliably completes the flip, indicating robustness to sensing and dynamics perturbations.

**(ii) Human–robot fruit handover.** A human hands fruit to the robot. The robot must execute a smooth, anticipatory interaction while synchronizing its timing with the human and remaining robust to unpredictable motions and sensing noise. This task is run on a Franka robot.

We collect a single demonstration to align the robot's motion phase with the human hand trajectory. At execution time, a pretrained CLIP model (Radford et al., 2021) provides a fruit-detection score, which we combine with the deviation from the demonstrated hand trajectory to define the distance field. This field determines the robot's phase and progression; the robot follows the progression flow until the desired phase is reached, yielding synchronized and fluid handovers. We observe 46 successful handovers out of 50 trials, resulting in a 92% success rate. A trial is counted as successful

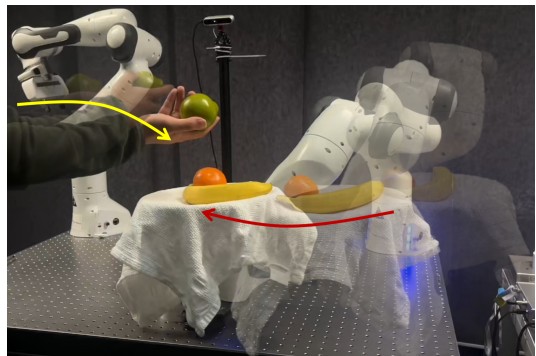

Figure 9: Real robot experiment on human-robot interaction task.

if the robot correctly recognizes the fruit and
anticipates and adapts to the human hand mo-
tion to complete the delivery. During execution,
we vary the object shapes and initial positions
while still using only a single demonstration, and the policy consistently completes the task, demon-
strating robustness and generalization.

More details about the robot platform, experimental setup, and training details are illustrated in
Appendices C.6 and C.7, respectively. The robot behavior during two tasks is shown in Figures 8, 9
and the attached video.

## 4 RELATED WORK

Among approaches to acquiring robotic skills—reinforcement learning (Sutton & Barto, 1998)
and optimal control (Bertsekas, 1995), imitation learning (IL) (Osa et al., 2018) stands out for not
requiring explicit task models or cost functions, making it especially appealing when dynamics
are hard to model. Even when such models exist, demonstrations can accelerate and improve
solutions (Nair et al., 2018; Razmjoo et al., 2021). Early approaches focus on time-dependent
dynamical movement primitives, such as Dynamic movement primitives (DMP) (Ijspeert et al.,
2013) and Probabilistic Movement Primitives (ProMP) (Paraschos et al., 2013), or time-independent
dynamical systems (Khansari-Zadeh & Billard, 2011). They provide well-established approaches
and efficient frameworks, but are usually limited in capturing complex, multi-modal demonstration
patterns. Recent learning-based approaches, such as Implicit Behavior cloning and Diffusion policy,
address this issue and have demonstrated impressive performance across a range of tasks (Florence
et al., 2022; Chi et al., 2023; Zhang & Gienger, 2024). However, these methods introduce challenges
such as hard to train, slow inference, and need multi-step inference (LeCun et al., 2006; Du &
Mordatch, 2019; Song & Ermon, 2019; Nijkamp et al., 2020; Zhang & Gienger, 2024). GPI bridges
dynamical systems and modern learning by representing demonstrations as distance fields—linking
naturally to metric learning for high-level scene representations while inducing flow fields for
low-level control. The closest prior, VINN (Pari et al., 2022), learns visual representations via
self-supervision and retrieves policies with $k$NN, achieving strong visual imitation. In contrast, GPI
supports diverse latent representations and synthesizes policy flows—demonstrating effectiveness
on tasks with complex dynamics.

## 5 LIMITATION AND CONCLUSION

We present Geometry-aware Policy Imitation (GPI), which treats demonstrations as geometric
curves that induce a distance field and policy flows. This perspective yields a simple, flexible,
efficient, multimodal, and interpretable policy that composes behaviors and integrates with diverse
latent representations. Our approach has a few limitations that are worth exploring in future work:

**Choice of distance metric and representation.** The metric and visual representation are the key
design levers that shape the induced flows. In this work, we rely on simple, manually specified met-
rics and off-the-shelf encoders. Making these components learnable and co-optimizing them with
policy synthesis—potentially conditioned on task or context—could further improve robustness and
out-of-distribution generalization while preserving the geometric structure that makes GPI inter-
pretable. Another promising direction is to leverage large models to provide task-relevant robotic
features (Intelligence et al., 2025; Barreiros et al., 2025).

**Scene dynamics and stability.** Our current results follow the standard imitation learning paradigm:
environment dynamics and unactuated components are treated as unknown, and policies are learned
purely from data rather than from a full dynamics model. A natural extension is to incorporate known
or learned dynamics models into the flow construction and analyze when the resulting closed loop
is provably stable and robust, for example via Lyapunov or contraction certificates with perturbation
and model-mismatch bounds. This could provide stronger guarantees in safety-critical settings.
Our Lyapunov-style analysis assumes a smooth distance field in the actuated subspace, so it does
not formally extend to discretized or non-smooth demonstrations, even though our time-discretized
benchmarks remain empirically robust.

**Scalability of demonstrations.** GPI stores only latent features and distances are computed in a
single batched operation, leading to favorable latency and memory scaling in our empirical study.

However, the memory footprint still grows linearly with the number of stored states. Future work could reduce this dependence via compact implicit distance parameterizations while preserving geometric fidelity and fast retrieval.

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

APPENDIX

## A CONVERGENCE OF THE FLOW POLICY

We prove convergence of the policy introduced in Section 2.1, which combines progression and attraction flows to form a stable dynamical system in the actuated subspace. For clarity, we rewrite the flow policy (equation 2) as

$$\dot{\boldsymbol{x}} = \lambda_1 \dot{\boldsymbol{x}}_{t^*} - \lambda_2 \nabla d(\boldsymbol{x}), \tag{7}$$

where $d(\boldsymbol{x})$ is the distance to the demonstration, $\nabla d(\boldsymbol{x})$ its gradient, $\dot{\boldsymbol{x}}_{t^*}$ the tangent velocity at the projection point $\boldsymbol{x}_{t^*}$, and $\lambda_1, \lambda_2 \geq 0$ weight progression and attraction.

We analyze stability using the Lyapunov function

$$V(\boldsymbol{x}) = \tfrac{1}{2} d^2(\boldsymbol{x}) \geq 0, \tag{8}$$

which vanishes only on the demonstration. Its time derivative is

$$\dot{V}(\boldsymbol{x}) = d(\boldsymbol{x}) \nabla d(\boldsymbol{x})^\top \dot{\boldsymbol{x}}. \tag{9}$$

Substituting the dynamics gives

$$\dot{V}(\boldsymbol{x}) = d(\boldsymbol{x}) \nabla d(\boldsymbol{x})^\top \left( \lambda_1 \dot{\boldsymbol{x}}_{t^*} - \lambda_2 \nabla d(\boldsymbol{x}) \right). \tag{10}$$

To simplify this expression, we use the fact that the projection point $\boldsymbol{x}_{t^*}$ is defined as the minimizer of the squared distance

$$\|\boldsymbol{x}_t - \boldsymbol{x}\|^2. \tag{11}$$

At this minimizer, the derivative with respect to $t$ must vanish:

$$(\boldsymbol{x}_{t^*} - \boldsymbol{x})^\top \dot{\boldsymbol{x}}_{t^*} = 0. \tag{12}$$

This condition implies that the displacement vector $\boldsymbol{x}_{t^*} - \boldsymbol{x}$, and therefore the gradient $\nabla d(\boldsymbol{x})$, is orthogonal to the trajectory tangent $\dot{\boldsymbol{x}}_{t^*}$:

$$\nabla d(\boldsymbol{x})^\top \dot{\boldsymbol{x}}_{t^*} = 0. \tag{13}$$

With this orthogonality property, the Lyapunov derivative reduces to

$$\dot{V}(\boldsymbol{x}) = -\lambda_2 d(\boldsymbol{x}) \|\nabla d(\boldsymbol{x})\|^2 \leq 0, \tag{14}$$

with equality only if $d(\boldsymbol{x}) = 0$. This shows that the system is globally stable and asymptotically converges to the demonstrated trajectory in the actuated space.

**Assumptions and scope.** This analysis is carried out for a continuous-time system in the actuated subspace, assuming a smooth demonstration trajectory and a well-defined, differentiable distance field around it. The result should therefore be interpreted as a convergence guarantee for this idealized setting. In practice, GPI is implemented in discrete time and with time-discretized demonstrations; the same flow construction is used, and we empirically observe stable rollouts in all benchmarks.

## B GPI ALGORITHM

---

**Algorithm 1** Geometry-Aware Policy Imitation

---

**Require:** $\mathcal{D} = \{\Gamma^{(i)}\}_{i=1}^{N}$, each $\Gamma^{(i)} = \{(\boldsymbol{x}_t^{(i)}, \boldsymbol{u}_t^{(i)})\}_{t=0}^{T_i}$; projection $P$; encoder $\Psi$; robot/environment distances $d_{\text{rob}}, d_{\text{env}}$; mixing $\alpha_{\text{rob}}, \alpha_{\text{env}} > 0$; weights $\lambda_1(\cdot), \lambda_2(\cdot)$; temperature $\beta$; top-$K$

**Ensure:** Control $\boldsymbol{u} \in \mathcal{X}'$ at query $\boldsymbol{x}_o$

1: $\boldsymbol{x}_o' \leftarrow P(\boldsymbol{x}_o), \quad \boldsymbol{z}_o \leftarrow \Psi(\boldsymbol{x}_o)$

2: **for all** $i \in \{1, \dots, N\}$ **(parallel over demonstrations) do**

3:     **Per-time distances**

$$\boldsymbol{d}_{\text{rob}}^{(i)} \leftarrow \big(d_{\text{rob}}(\boldsymbol{x}_o', \boldsymbol{x}_t'^{(i)})\big)_t, \quad \boldsymbol{d}_{\text{env}}^{(i)} \leftarrow \big(d_{\text{env}}(\boldsymbol{z}_o, \Psi(\boldsymbol{x}_t^{(i)}))\big)_t$$

4:     **Combined distance:** $\quad \boldsymbol{d}^{(i)} \leftarrow \alpha_{\text{rob}} \boldsymbol{d}_{\text{rob}}^{(i)} + \alpha_{\text{env}} \boldsymbol{d}_{\text{env}}^{(i)}$

5:     **Nearest time index and scalar distance:**

$$\kappa^{(i)}(\boldsymbol{x}_o) \leftarrow \arg\min_t \boldsymbol{d}_t^{(i)}, \qquad d(\boldsymbol{x}_o \mid \Gamma^{(i)}) \leftarrow \min_t \boldsymbol{d}_t^{(i)}$$

6:     **Progression flow:** $\quad \boldsymbol{u}_\kappa^{(i)} \leftarrow \boldsymbol{u}_{\kappa^{(i)}(\boldsymbol{x}_o)}^{(i)} = \dot{\boldsymbol{x}}_{\kappa^{(i)}(\boldsymbol{x}_o)}'^{(i)}$

7:     **Attraction flow:** $\quad \boldsymbol{u}_{\text{att}}^{(i)} \leftarrow -\nabla_{\boldsymbol{x}_o'} d_{\text{rob}}\big(\boldsymbol{x}_o', \boldsymbol{x}_{\kappa^{(i)}(\boldsymbol{x}_o)}'^{(i)}\big)$

8:     **Local policy:**

$$\pi_i(\boldsymbol{x}_o) \leftarrow \lambda_1\big(d(\boldsymbol{x}_o \mid \Gamma^{(i)})\big) \boldsymbol{u}_\kappa^{(i)} + \lambda_2\big(d(\boldsymbol{x}_o \mid \Gamma^{(i)})\big) \boldsymbol{u}_{\text{att}}^{(i)}$$

9: **Top-$K$ selection by demonstration distance:** $\quad I_K \leftarrow$ indices of the $K$ smallest $d(\boldsymbol{x}_o \mid \Gamma^{(i)})$

10: **Softmax weights over selected demos:** $\quad w_i(\boldsymbol{x}_o) \leftarrow \dfrac{\exp\big(-\beta d(\boldsymbol{x}_o \mid \Gamma^{(i)})\big)}{\sum_{j \in I_K} \exp\big(-\beta d(\boldsymbol{x}_o \mid \Gamma^{(j)})\big)} \quad (i \in I_K)$

11: **Global policy:** $\quad \boldsymbol{u} = \pi(\boldsymbol{x}_o) = \displaystyle\sum_{i \in I_K} w_i(\boldsymbol{x}_o) \pi_i(\boldsymbol{x}_o)$

12: **return** $\boldsymbol{u}$

---

## C  IMPLEMENTATION DETAILS

### C.1  PUSHT TASK WITH STATE-BASED INPUTS

For low-dimensional states, each demonstration is represented as

$$\boldsymbol{x}_t^{(i)} = [x_a, y_a, x_b, y_b, \theta_b] \in \mathbb{R}^5,$$

where $(x_a, y_a)$ denote the agent position, $(x_b, y_b)$ the block position, and $\theta_b$ the block orientation. The associated action specifies the target location for a low-level controller:

$$\boldsymbol{u}_t^{(i)} = [x_{\text{target}}, y_{\text{target}}],$$

which we rewrite for velocity control as the relative displacement:

$$\boldsymbol{u}_t^{(i)} = [x_{\text{target}} - x_a, \ y_{\text{target}} - y_a].$$

All state variables are normalized to $[0, 1]$ before computing distances. The distance field $d(\boldsymbol{x}, \Gamma^{(i)})$ is defined as the weighted sum of three components:

$$d(\boldsymbol{x}, \boldsymbol{x}_t^{(i)}) = w_{\text{obj}} \|(x_b, y_b) - (x_b^{(i)}, y_b^{(i)})\|_2 + w_{\text{agt}} \|(x_a, y_a) - (x_a^{(i)}, y_a^{(i)})\|_2 + w_\theta \, \text{ang}(\theta_b, \theta_b^{(i)}), \quad (15)$$

where $\text{ang}(\cdot, \cdot)$ denotes angular distance. Unless otherwise stated, the weights are set to $w_{\text{obj}} = w_{\text{agt}} = w_\theta = 1.0$.

Each demonstration induces a distance field and an associated flow policy. At inference time, the global policy is formed by composing the $K$ nearest demonstration policies, with $\lambda_1 = \lambda_2 = 1.0$. Evaluation is performed on environment seeds 500–510 using three distinct policy seeds.

We further explore several variants to improve the flexibility of GPI:

**Relative vs. absolute state representation.** The PushT task involves nonlinear contact dynamics, so the choice of state representation is important. In the *relative* variant, the agent position is expressed in the object's coordinate frame:

$$\tilde{\boldsymbol{p}}_a = R(-\theta_b) \left( (x_a, y_a) - (x_b, y_b) \right), \quad (16)$$

where $R(-\theta_b)$ is the SE(2) rotation matrix aligning the block's orientation to the $x$-axis. The demonstrated action $\boldsymbol{u}_t$ is similarly transformed. During execution, the predicted action $\tilde{\boldsymbol{u}}$ is mapped back to global coordinates via the inverse transformation:

$$\boldsymbol{u} = R(\theta_b) \, \tilde{\boldsymbol{u}} + (x_b, y_b). \quad (17)$$

**Smooth flow fields.** When the action horizon is set to 1, the controller is highly reactive and may produce abrupt changes whenever the nearest demonstration switches. To mitigate this, we apply first-order smoothing to the action sequence:

$$\boldsymbol{u}_t^{\text{smooth}} = \alpha \, \boldsymbol{u}_t + (1 - \alpha) \, \boldsymbol{u}_{t-1}^{\text{smooth}}, \quad (18)$$

where $\alpha \in [0, 1]$ is a smoothing parameter.

**Recent-action suppression.** To mitigate oscillatory behavior arising from repeatedly selecting near-identical actions, we maintain a sliding-window memory $\mathcal{M}$ of the most recent $M$ actions. During action selection, if the candidate $\boldsymbol{u}_t$ lies within a tolerance $\epsilon$ of any element in $\mathcal{M}$, it is suppressed and the next-best candidate from the composed policy is chosen. This mechanism enforces diversity over short horizons, prevents immediate backtracking to previously executed actions, and ensures the policy explores novel trajectories while preserving responsiveness.

**Perturbed query states.** To evaluate robustness, we perturb the query agent position with additive Gaussian noise:

$$\tilde{\boldsymbol{x}}' = \boldsymbol{x}' + \epsilon, \qquad \epsilon \sim \mathcal{N}(0, \sigma^2 I), \quad (19)$$

where $\boldsymbol{x}' = (x_a, y_a)$ is the agent substate. The noise variance $\sigma^2$ is annealed over time, decaying from $\sigma = 0.1$ at the beginning of execution to $\sigma = 0.001$ at later steps. This perturbation injects stochasticity into the query states, which increases variability in the retrieved flows and can induce multimodal behaviors.

**Subsampled demonstrations.** For efficiency and robustness, instead of using all demonstrations, we randomly sample a subset $\Gamma_{\text{sub}} \subset \Gamma$ at each query. The global policy is then composed over $\Gamma_{\text{sub}}$. Empirically, we find that subsampling does not reduce performance; in some cases, the induced stochasticity even helps the agent escape undesirable cycles or "stacked" behaviors.

## C.2 PUSHT TASK WITH VISION-BASED INPUTS

In the PushT environment, observations consist of an RGB image $\mathbf{I}$ together with agent positions $(x_a, y_a)$. Each demonstration state is represented as

$$\boldsymbol{x}_t^{(i)} = [\, x_a, y_a, \mathbf{I}\,].$$

**Vision encoder.** To obtain compact image features, we use an encoder $\psi$ with a ResNet-18 backbone (group normalization) and a projection head (MLP with sizes [512, 256, 128, 3]). The encoder is trained with a mean squared error (MSE) loss to predict the object position and orientation:

$$\psi(\mathbf{I}) \approx [x_o, y_o, \theta_o], \qquad \mathcal{L}_{\text{MSE}} = \tfrac{1}{B} \sum_{i=1}^{B} \big\| \boldsymbol{x}_{\text{pred}}^{(i)} - \boldsymbol{x}_{\text{target}}^{(i)} \big\|_2^2.$$

Training is performed for 200 epochs using the Adam optimizer with a learning rate of 0.001.

**Distance metric and policy synthesis.** After training, each demonstration image is embedded as

$$\boldsymbol{z}_t^{(i)} = \psi(\mathbf{I}_t^{(i)}),$$

and for a query state $\boldsymbol{x}_o = [x_a, y_a, \mathbf{I}]$,

$$\boldsymbol{z}_o = \psi(\mathbf{I}).$$

Distances are defined in this learned feature space and policy synthesis then proceeds identically to the state-based inputs.

## C.3 PUSHT TASK WITH RESNET-18 ENCODER AND PCA

We construct a compact observation embedding by reusing the same ResNet-18 encoder from the Diffusion Policy implementation (task-pretrained on PUSHT). At inference, this encoder is frozen and used as a fixed feature extractor. We aggregate features over a short temporal window (`obs_horizon = 2`), apply PCA for dimensionality reduction on the image features, and concatenate with the last two agent positions (normalized and reweighted to balance scale). Each demonstration is thus represented in this joint embedding space. At test time, the current observation is embedded in the same way, and the closest demonstration under cosine similarity is identified. The policy then follows the flow induced by this demonstration, with progression and attraction weights set to $\lambda_1 = \lambda_2 = 1.0$.

**Per-timestep features.** Given an image $\boldsymbol{I}$ and agent position $[x_a, y_a]$, we extract a 512-D descriptor $\psi(\boldsymbol{I})$ with the frozen ResNet-18 backbone (final FC removed; BatchNorm $\rightarrow$ GroupNorm as in the diffusion policy).

**Temporal windowing and dimensionality reduction.** With `obs_horizon` $T = 2$, we flatten the last $T$ descriptors and apply IncrementalPCA to project them to 16 principal components:

$$\boldsymbol{z}_t \;=\; \text{PCA}_{16}([\psi(I_{t-1}), \psi(I_t)]) \in \mathbb{R}^{16}.$$

**Concatenation with agent positions.** To balance image and agent information, we concatenate the PCA embedding $z_t$ with the normalized agent positions from the last two steps. All embeddings are L2-normalized before similarity computations.

**Policy selection.** At test time, the query embedding is compared to the demonstration database using cosine similarity, and the flow is executed with $\lambda_1 = \lambda_2 = 1.0$. To prevent degenerate repeats, the selected pair is removed from the database at the next step.

## C.4 PUSHT TASK WITH VAE

We construct a compact observation embedding using a convolutional variational autoencoder (VAE) trained directly on PUSHT images. At inference, we discard the decoder and use only the encoder to produce latent codes, which are concatenated with scaled agent positions to form the final embedding. The global policy then follows the flow induced by the closest demonstration under cosine similarity, with progression and attraction weights set to $\lambda_1 = \lambda_2 = 1.0$.

**Per-timestep features.** Given an image $I_t$ with pixel values normalized to $[0, 1]$, the VAE encoder outputs a Gaussian posterior

$$z_t \sim q_\phi(z \mid I_t), \quad z_t \in \mathbb{R}^d,$$

with diagonal covariance. At inference, we use only the posterior mean $\mu_t$ as the latent feature.

**Retrieval.** At test time, we encode the current observation window to obtain $z_t$, normalize it, and compute cosine similarity against the stored database features. The demonstration with the highest similarity is selected, and its associated action sequence defines the flow. Cosine similarity achieved slightly higher performance (average return $\approx 0.88$) compared to Euclidean distance ($\approx 0.85$).

**Training Setup.** We train the VAE with a standard Gaussian prior $p(\mathbf{z}) = \mathcal{N}(\mathbf{0}, I)$ and a Gaussian reconstruction likelihood $p(\mathbf{x} \mid \mathbf{z}) = \mathcal{N}(\hat{\mathbf{x}}(\mathbf{z}), \tau^2 I)$ with fixed $\tau = 2 \times 10^{-1}$. This choice of $\tau$ balanced the reconstruction and KL terms: with $\tau = 0.2$ both the reconstruction loss and the KL divergence decreased steadily, whereas using smaller $\tau$ values led to optimization stalling (neither term decreased). Training was performed for 25 epochs with the Adam optimizer (learning rate $1 \times 10^{-4}$). At inference, we discard the decoder and use only the encoder's posterior mean.

### C.5 PUSHT TASK WITH SAM-BASED POSE EMBEDDING

We estimate object pose directly from images using a pretrained SAM/SAM2 pipeline (no fine-tuning). From each frame we obtain a binary mask of the T-block, from which we extract its centroid $(x_b, y_b)$ and axial orientation $\theta_b$ (defined modulo $\pi$). Combined with the agent position $(x_a, y_a)$, this yields the state

$$\boldsymbol{x}_t = [\, x_a, y_a, x_b, y_b, \theta_b \,] \in \mathbb{R}^5.$$

All variables are normalized to $[0, 1]$ before distance computations; angular differences use the same axial angular distance as in the state-based setup. Distances and policy composition follow the same formulation, with weights $w_{\text{obj}} = w_{\text{agt}} = w_\theta = 1.0$ and flow execution with $\lambda_1 = \lambda_2 = 1.0$.

**Per-timestep pose extraction.** Given a SAM mask, the centroid is

$$(x_b, y_b) = \text{centroid}(\text{mask}),$$

and the orientation is computed from second-order moments of foreground pixels. Let $\mu_{pq}$ denote centralized moments; the principal axis corresponding to the largest covariance eigenvalue indicates the elongation direction. We define

$$\theta_b = \tfrac{1}{2} \operatorname{atan2}\big(2\mu_{11}, \mu_{20} - \mu_{02} + \varepsilon\big),$$

wrap $\theta_b$ to $(-\pi, \pi]$, and treat it as axial (modulo $\pi$) for angular distance.

**Retrieval and policy selection.** At test time, we form $\boldsymbol{x}_t = [x_a, y_a, x_b, y_b, \theta_b]$, apply the same normalization as above, and compute distances to all stored demonstration states using the state-based metric. We retrieve the $K$ nearest neighbors (default $K = 1$) and execute the composed flow with $\lambda_1 = \lambda_2 = 1.0$.

**Tracking and prompting details.** We use SAM2's video predictor (`sam2.1_hiera_tiny`) to track the T-block across frames, re-prompting each step with a skeletal outline derived from the most recent pose estimate to stabilize mask propagation. To compensate for a small systematic bias in predicted centroids, we apply a constant offset correction to $(x_b, y_b)$, calibrated on seeds 500–700.

**Limitations.** Performance depends on segmentation quality; occlusions and viewpoint changes can induce drift in the estimated pose, which in turn affects retrieval and control.

### C.6 ROBOT-FLIP TASK

**Robot teleoperation**: We utilized a bimanual robotic system configured with a ViperX300s (follower) and a WidowX250 (leader), along with a RealSense D405 camera from a top-down view. The system is built on an open-source platform. By using robot teleoperation, we collected 121 demonstrations, each contains 200 to 1000 timesteps to complete the flip task. The dataset is structured in an HDF5 format and includes robot actions and observations, where observations are composed of effort, images, joint angles, and joint velocities. Specifically, we teleoperated the leader robot (WidowX250) to control the follower (ViperX300s) robot for manipulation tasks (flip the box). The

camera records images at an 848×480 resolution with a 30 Hz frequency, and then crops them to a 320×240 resolution for policy training.

**Policy imitation.** The policy imitation process is similar to the pushT task with vision-based inputs. Specifically, we use a vision encoder that takes RGB images as input and predicts the desired robot action as a latent embedding using an MSE loss. Training is performed for 100 epochs using the Adam optimizer with a learning rate of 0.0001. After training, we calculate the latent feature of each demonstrated image as a feature database. The online inference involves the computation of a distance field that includes both distance measurement in this latent space and an additional distance metric for joint position displacement, guiding the flow field and policy composition. Both attraction and progression parameters are set to 1.0 during execution. To ensure the temporal consistency, the task is run with horizon=100.

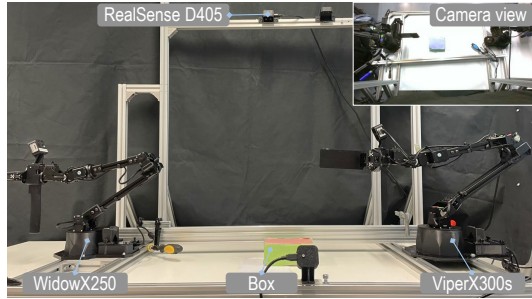

Figure 10: ALOHA teleoperation platform.

## C.7 HUMAN–ROBOT INTERACTION TASK

We use the `openai/clip-vit-base-patch32` CLIP model for vision–language grounding. Positive and negative text prompts for hand–held object detection are listed below.

**Text prompts.**

```
pos_prompts = [
    "a photo of a hand holding a banana",
    "a hand holding an apple",
    "a human hand holding an orange",
    "a hand holding a pear",
    "a hand holding a strawberry",
    "a hand holding grapes",
    "a hand holding a piece of fruit",
    "a person's hand holding a fruit",
    "close-up of a hand holding a fruit",
]

neg_prompts = [
    "an empty hand",
    "a hand with nothing in it",
    "a hand holding a baseball",
    "a hand holding a black ball",
    "a hand holding a blue cup",
    "a hand holding a plastic cup",
    "a hand holding adhesive tape",
    "a hand holding a tape roll",
    "a hand holding a screwdriver",
    "a hand holding a tool",
    "a hand holding a non-fruit object",
]
```

# D    ADDITIONAL EXPERIMENTAL RESULTS

## D.1    MEMORY COST

The state-based PUSHT dataset has $25,000 \times 7 = 175,000$ elements, requiring $175,000 \times 4 \approx 0.67$ MB with `float32`, consistent with the observed 0.7 MB. For comparison, an MLP with layers $[7, 512, 256, 128, 1]$ has 168,449 parameters ($\approx 0.64$ MB), which is at a similar scale. However, typical models are far larger than simple MLP; e.g., a state-based diffusion policy exceeds 200 MB.

Although GPI's memory grows linearly with the number of demonstrations, this is practical in our setting: robot actions are low-dimensional, and high-dimensional observations are stored as compact latent features. Inference is lightweight, parallelizable, and can use subsampling or approximate nearest-neighbor search to bound latency. As we demonstrated in the paper, GPI achieves orders-of-magnitude gains in efficiency over standard baselines in common imitation-learning settings.

## D.2    COMPLEXITY AND SCALABILITY

**Complexity of one control step.** All demonstrations are stored in a tensor of shape $(NT, D)$, where $N$ is the number of demonstrations, $T$ is the trajectory length, and $D$ is the state / feature dimension. Given the current observation of shape $(1, D)$, we compute its distance to all stored states in a single batched operation. The complexity of this retrieval step is therefore $O(NTD)$, implemented in parallel on GPU. Once the top-$K$ neighbors are selected, combining their flows to compute the final control command is $O(KD)$, which is negligible for small $K$ compared to the retrieval cost.

**Empirical scalability.** To make the scaling explicit, we report retrieval latency and memory usage as we vary the number of stored states from $10^2$ to $10^6$ and the feature dimension $D$ from 5 to 512. Latency is measured in milliseconds (ms) on a single GPU, and memory usage is reported in megabytes (MB). We measure the cost of computing Euclidean distances between a single query state and all stored features, which is the component that grows with the database size:

Table 4: Retrieval latency and memory usage for a single query as a function of the number of stored states and feature dimension $D$. Each entry reports *latency (ms) / memory (MB)*.

| # states | $D = 5$ | $D = 32$ | $D = 128$ | $D = 512$ |
|---|---|---|---|---|
| $1 \times 10^2$ | 0.039 / 0.00 | 0.033 / 0.01 | 0.034 / 0.05 | 0.033 / 0.20 |
| $1 \times 10^3$ | 0.039 / 0.02 | 0.036 / 0.12 | 0.035 / 0.49 | 0.039 / 1.96 |
| $1 \times 10^4$ | 0.033 / 0.19 | 0.037 / 1.22 | 0.068 / 4.88 | 0.217 / 19.6 |
| $1 \times 10^5$ | 0.051 / 1.91 | 0.221 / 12.2 | 0.517 / 48.8 | 1.973 / 196 |
| $1 \times 10^6$ | 0.291 / 19.1 | 1.920 / 122 | 4.891 / 488 | 19.26 / 1955 |

Even with very large databases ($10^6$ states) and high-dimensional features ($D = 512$), retrieval remains below 20 ms with about 2 GB (approximately 1955 MB) of memory, which is compatible with typical real-time manipulation settings.

## D.3    ABLATIONS ON DISTANCE METRICS

In all experiments, the metric used by GPI is constructed from simple and physically motivated components: Euclidean distance for robot states (e.g., joint angles, end-effector positions), geodesic distance for quaternions, and cosine similarity for latent visual embeddings. These choices follow the geometry and physical meaning of each state component. All state dimensions are normalized before distance computation, which further reduces sensitivity to manual tuning.

To assess sensitivity to the specific form of the metric, we perform an ablation on the PushT task comparing L1 distance, L2 distance, and cosine similarity, both in the original state space and in a VAE-based latent space. The average rewards are summarized in Table 5.

In the state space, both L1 and L2 distances perform well, indicating that GPI is not sensitive to the exact choice of norm as long as the metric is consistent with the underlying state geometry. In the latent space, cosine similarity performs best, with L1 and L2 still competitive, consistent with

Table 5: Average reward for different distance metrics on the pushT task.

| Metric | State space | Latent space (VAE) |
|---|---|---|
| L1 norm distance | 88 | 81 |
| L2 norm distance | 86 | 85 |
| Cosine similarity | 88 | 87 |

standard practice for feature embeddings. Even when directly using cosine similarity in the state space, performance remains reasonable. Overall, these results indicate that GPI does not rely on fragile, hand-tuned metrics and works reliably with simple, task-aligned distances.

We also study how the relative weighting between the environment distance $d_{\text{env}}$ and the robot distance $d_{\text{rob}}$ influences performance on PushT and Robomimic tasks. We define the combined metric

$$d = w_{\text{rob}}\, d_{\text{rob}} + w_{\text{env}}\, d_{\text{env}}, \tag{20}$$

and vary the ratio $w_{\text{env}}/w_{\text{rob}}$. The results are given in Table 6. Performance is clearly degraded when the environment weight is extremely small or large (e.g., $w_{\text{env}}/w_{\text{rob}} = 0.01$ or $100$), but remains high over a broad intermediate range (approximately $w_{\text{env}}/w_{\text{rob}} \in [0.1, 10]$), with the best results typically obtained near equal weighting (around 0.5–1.0). These trends confirm that the metric and its weighting are important design choices, but also show that GPI remains robust to a wide range of relative weightings between robot and environment distances.

Table 6: Evaluation of robot–environment distance weighting on various benchmarks.

| $w_{\text{env}}/w_{\text{rob}}$ | 0.01 | 0.1 | 0.5 | 1.0 | 5.0 | 10.0 | 100.0 |
|---|---|---|---|---|---|---|---|
| PushT | 0.38 | 0.76 | 0.83 | 0.87 | 0.80 | 0.80 | 0.54 |
| Lift | 0.68 | 0.85 | 0.98 | 1.00 | 0.83 | 0.78 | 0.52 |
| Can | 0.61 | 0.73 | 0.88 | 0.96 | 0.79 | 0.58 | 0.18 |
| Square | 0.23 | 0.55 | 0.72 | 0.82 | 0.63 | 0.51 | 0.16 |

### D.4  ABLATION ON COMPOSITION HYPERPARAMETERS

We ablate the softmax temperature $\beta$ used in (3) for composing the top-$K$ demonstrations on PushT (state-based). As shown in Table 7, performance remains stable over several orders of magnitude of $\beta$.

Table 7: Effect of the softmax temperature $\beta$ on PushT.

| $\beta$ | $10^{-4}$ | $10^{-3}$ | $10^{-2}$ | $10^{-1}$ |
|---|---|---|---|---|
| Avg. Score | $85.1 \pm 1.2$ | $84.8 \pm 0.9$ | $85.8 \pm 1.2$ | $85.6 \pm 1.4$ |

### D.5  ROBOMIMIC AND ADROIT HAND TASKS

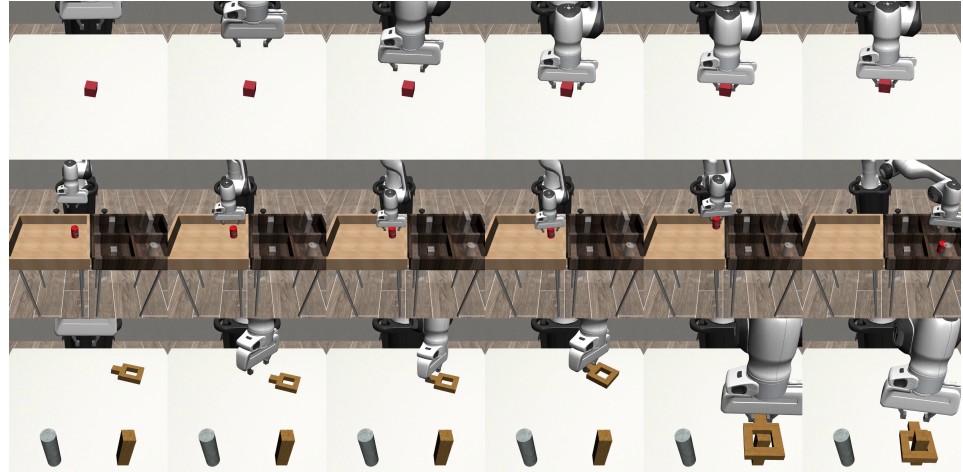

Figure 11: Snapshots of experimental results for Lift, Can, and Square tasks on Robomimic environments.

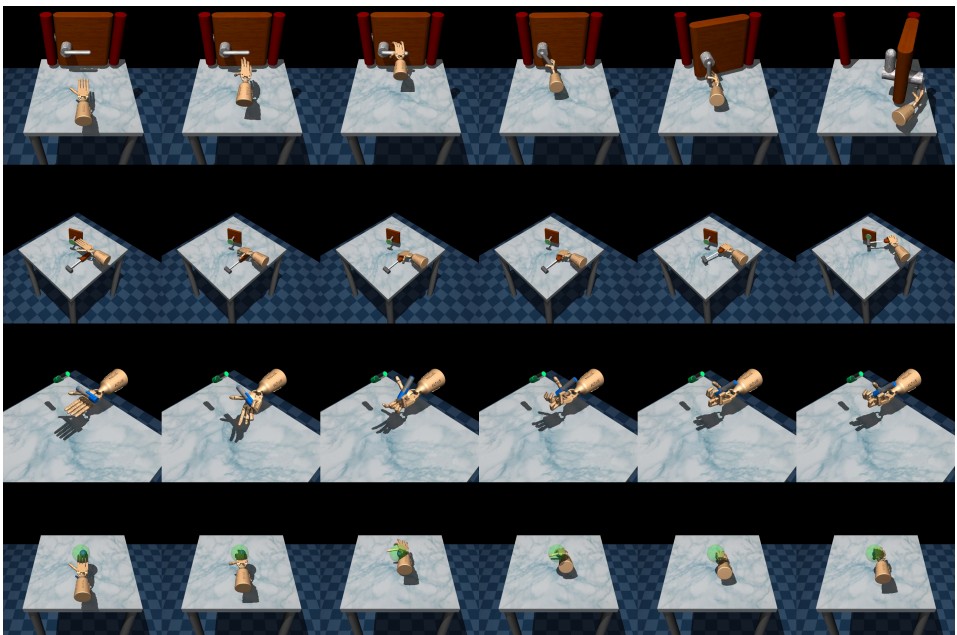

Figure 12: Snapshots of experimental results for Door, Hammer, Pen, and Relocate on Adroit hand tasks.

### D.6  2D MAZE

We evaluate our approach on the 2D Maze benchmark, previously used by (Chen et al., 2025; Janner et al., 2022). Unlike these methods, our approach is *training-free*: at test time we select a suffix of a single demonstration using a simple distance metric and execute it. Concretely, for demonstration $i$ of length $H$ and timestep $k$, we minimize

$$D(i, k) = 10 \, \|\mathbf{x}_0 - \mathbf{x}_k^{(i)}\|_2 + 5 \, \|\mathbf{x}_g - \mathbf{x}_g^{(i)}\|_2 + 0.1 \, (H - k),$$

where $\mathbf{x}_0$ is the initial state, $\mathbf{x}_k^{(i)}$ is the $k$-th state of demonstration $i$, $\mathbf{x}_g$ is the task goal, and $\mathbf{x}_g^{(i)}$ is the goal state associated with demonstration $i$. The final term penalizes long remaining horizons; since 2D Maze demonstrations can include detours, this bias favors suffixes that proceed more directly to the goal. After selecting $(i^\star, k^\star)$, we execute the suffix $\{\mathbf{x}_{k^\star:H}^{(i^\star)}\}$ as the plan. In doing so, our method also recovers the effective task horizon $H - k^\star$, something most alternative approaches cannot determine directly. Instead, they must either: (i) assume a long horizon and truncate once

the task is completed, (ii) assume a short horizon and repeat until completion, or (iii) try multiple horizons and select the smallest successful one.

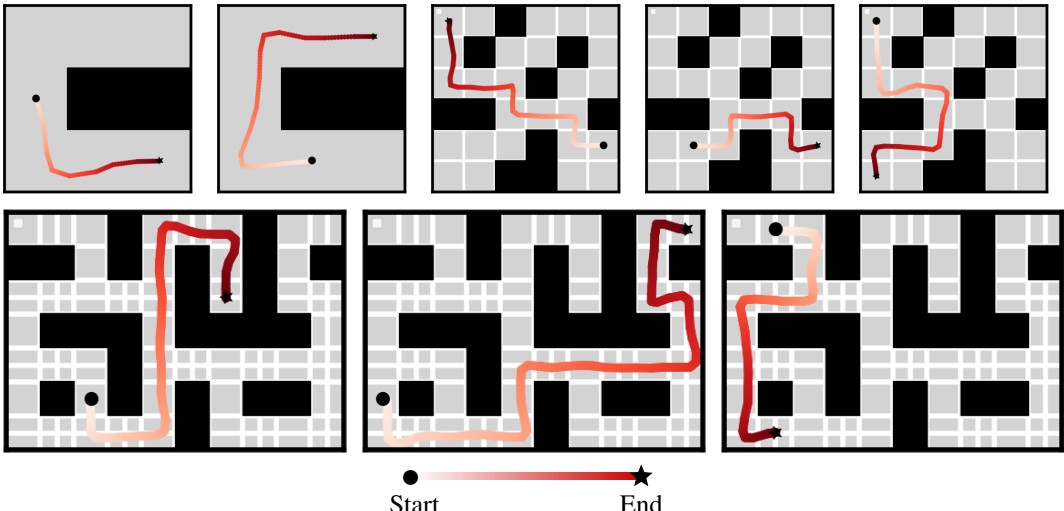

Figure 13: Results on 2D Maze using our method. Without any training, a simple distance-based criterion achieves a 100% success rate across all tasks, with an average inference time of 0.08 seconds.

## E    REPRODUCIBILITY STATEMENT

We will release our code, configuration files, and evaluation scripts upon publication. Key implementation details and protocols are documented in the main text and appendix to facilitate reproduction in the interim.

## F    USE OF LARGE LANGUAGE MODELS (LLMS)

We used LLMs (e.g., ChatGPT and Claude) to rephrase and polish the manuscript and to assist with coding tasks. All LLM-generated code was reviewed, edited, and integrated by the authors; the LLM did not design algorithms or produce experimental results.

