# OpenReview forum: "Geometry-aware Policy Imitation"
_ICLR.cc/2026/Conference — ICLR 2026 Poster_

### Official Review · Reviewer_tvdn · 2025-10-30

**Soundness:** 3
**Presentation:** 3
**Contribution:** 3
**Rating:** 8
**Confidence:** 3

**Summary:**

This paper proposes Geometry-Aware Policy Imitation (GPI), an imitation learning framework using expert demonstrations as state-space geometric curves. It generates a non-parametric vector field via distance fields, decouples metric learning and policy synthesis, and outperforms diffusion-based policies.

**Strengths:**

- This paper is overall very clearly written.

- The proposed method is straightforward in implementation yet very effective.

- Real-world experiments are valuable. I appreciate the authors' efforts to conduct real-world validations of their approach

**Weaknesses:**

**Weakness 1. Analysis of Stability Under Strong Perturbations**

While the paper demonstrates robustness to small Gaussian perturbations and unknown dynamics, it lacks evaluation under strong, real-world perturbations (e.g., sudden object collisions, sensor noise bursts, or large deviations from demonstration trajectories). The Lyapunov stability proof in Appendix A assumes continuous state/action spaces and ideal distance field computation, but no experiments test whether GPI maintains convergence when perturbations violate these assumptions. This gap raises doubts about GPI’s practicality for unstructured environments where severe perturbations are common.


**Weakness 2 Distance Metric Design and Adaptability**

The choice of distance metric (decomposed into $\(d_{rob}\) $and $\(d_{env}\)$) is a critical design lever for GPI, yet the paper relies on hand-tuned weights (e.g., $\(w_{obj}=w_{agt}=w_{\theta}=1.0\$) in PushT) and preselected metric types (Euclidean for positions, geodesic for quaternions) without systematic validation of their optimality. Additionally, while the paper mentions making the metric "learnable and co-optimized with policy synthesis" as future work, it provides no analysis of how suboptimal metric choices (e.g., misweighted \(d_{rob}\) vs. \(d_{env}\)) impact performance—especially in tasks with complex state spaces (e.g., high-dimensional vision + multi-joint robot states).

**Questions:**

see weakness section

---

> ### Author Response · Authors · 2025-11-21
> **Response to Reviewer tvdn**
>
> Dear Reviewer tvdn,
>
> Thank you for carefully reading our paper and for your thoughtful comments. Below we provide clarifications and additional results related to your concerns.
>
> ***Weakness 1. Analysis of Stability Under Strong Perturbations: While the paper demonstrates robustness to small Gaussian perturbations and unknown dynamics, it lacks evaluation under strong, real-world perturbations (e.g., sudden object collisions, sensor noise bursts, or large deviations from demonstration trajectories). The Lyapunov stability proof in Appendix A assumes continuous state/action spaces and ideal distance field computation, but no experiments test whether GPI maintains convergence when perturbations violate these assumptions. This gap raises doubts about GPI’s practicality for unstructured environments where severe perturbations are common.***
>
> We thank the reviewer for this insightful comment. The scope and limitations of our Lyapunov-based analysis are discussed in the **General Response to All Reviewers (R1)**, where we explicitly acknowledge the assumptions on smooth distance fields and actuated-subspace dynamics, and that the proof does not directly cover severe perturbations or unknown environment dynamics.
>
> On the empirical side, Figure 6 in our manuscript examines robustness to injected Gaussian noise in the state and shows how performance degrades as the noise level ranges from small to relatively strong. In the real-robot ALOHA box-flip task, we also introduce realistic disturbances such as clutter, partial occlusions, and external pushes (Fig. 8, bottom row, and the attached video); GPI continues to perform well in these settings, suggesting practical robustness in the scenarios we tested.
>
> Large deviations from the demonstration manifold and severe perturbations are a known out-of-distribution challenge for data-driven imitation methods. In this context, GPI offers two structural advantages: (1) when the assumptions of the analysis are reasonably satisfied, the induced dynamical system is provably attracted toward the demonstration curves in the actuated space; and (2) when they are violated, the policy still follows an explicit flow field constructed from the most similar demonstrations, rather than extrapolating arbitrary actions from a black-box network, which tends to yield more interpretable and predictable behavior under distribution shift. We view a dedicated robustness study under extreme perturbations as a promising direction for future work.
>
> ***Weakness 2. Distance Metric Design and Adaptability: The choice of distance metric (decomposed into $d_{\text{rob}}$ and $d_{\text{env}}$) is a critical design lever for GPI, yet the paper relies on hand-tuned weights and preselected metric types (Euclidean for positions, geodesic for quaternions) without systematic validation of their optimality. Additionally, while the paper mentions making the metric "learnable and co-optimized with policy synthesis" as future work, it provides no analysis of how suboptimal metric choices (e.g., misweighted $d_{\text{rob}}$ vs. $d_{\text{env}}$) impact performance—especially in tasks with complex state spaces (e.g., high-dimensional vision + multi-joint robot states).***
>
> We appreciate this observation. In the **General Response to All Reviewers(R2)**, we provide a detailed discussion and new ablations that vary both the metric type (L1, L2, cosine, state vs. latent space) and the relative weighting between $d_{\text{rob}}$ and $d_{\text{env}}$. These results show that GPI is empirically robust across a broad range of metric choices and weight ratios.
>
> We would like to emphasize two conceptual points that align with your concern:
>
> 1. Because the metric encodes task-dependent geometric structure, it is natural and important to choose or learn it in a principled and reasonable way.
> 2. In practice, the associated weights between  $d_{\text{rob}}$ and $d_{\text{env}}$ are relatively easy to tune: they are low-dimensional and interpretable, and our experiments indicate that coarse sweeps over a small set of values suffice, which is simpler than training and regularizing end-to-end parametric policy networks.
>
> We agree that making the metric learnable and co-optimizing it with policy synthesis, especially in high-dimensional visuomotor settings, is a promising extension. The additional experiments in R2 are a first step toward understanding metric sensitivity, and we see jointly learned, task-adaptive metrics as an exciting direction for future work building on the GPI framework.
>
> ---
> Finally, we thank you again for your positive assessment and for highlighting important issues around robustness and metric design. We believe the clarified analysis and new experiments substantially strengthen the paper.

---

### Official Review · Reviewer_aXzo · 2025-11-01

**Soundness:** 3
**Presentation:** 3
**Contribution:** 4
**Rating:** 6
**Confidence:** 3

**Summary:**

This paper proposes Geometry-aware Policy Imitation (GPI), a non-parametric approach to imitation learning that treats demonstrations as geometric curves inducing distance fields in state space. From these fields, GPI derives two control primitives: a progression flow (advancing along trajectories) and an attraction flow (correcting deviations). The method achieves competitive or superior performance compared to diffusion policies while being 20-100× faster and requiring less memory. Experiments span simulation (PushT, RoboMimic, Adroit) and real robots (ALOHA, Franka), demonstrating efficiency, multimodality, and robustness.

**Strengths:**

- The geometric perspective on imitation learning is intuitive and well-motivated. The idea is quite novel and interesting.
- The policy inference speedup and memory reduction have a significant advantage over Diffusion Policy.
- The 2D example is illustrative.

**Weaknesses:**

- The policy rollout success rates is almost the same as Diffusion Policy. The authors may use more complex tasks to show the performance differences.
- The real robot experiments are limited and do not report success rates.
- The paper does not compare to VINN (Pari et al. 2022), though cited as the closest prior work.
- The paper does not explore under what conditions the convergence proof holds when dynamics are unknown/misspecified.
- The paper does not show performances when demonstrations are suboptimal or contain mistakes.

**Questions:**

See weaknesses.

---

> ### Author Response · Authors · 2025-11-21
> **Response to Reviewer aXzo**
>
> Dear Reviewer aXzo,
>
> Thank you for your valuable comments. Our clarification on the convergence analysis and its assumptions is provided in the  **General Response to All Reviewers (R1)**. Below, we address points that are specific to your review.
>
> ***The policy rollout success rates are almost the same as Diffusion Policy. The authors may use more complex tasks to show the performance differences.***
>
> We thank the reviewer for this suggestion. Diffusion Policy is a well-established and competitive baseline, so achieving similar success rates on standard benchmarks is a meaningful result, especially given that GPI is a non-parametric geometric method with substantially lower training cost, lighter memory footprint, and faster inference. Across our benchmarks, GPI attains comparable or higher success rates than Diffusion Policy (and recent flow-matching policies, see Table 1 in the revised manuscript) on most tasks, while additionally providing a simpler and more interpretable policy structure.
>
> Regarding “more complex” tasks, we would like to clarify that our fruit-delivery experiment targets a complementary challenge: by leveraging pretrained vision–language models, GPI completes the task from **only a single demonstration**, a regime that is typically difficult for diffusion- or flow-matching–based methods, which usually rely on larger training datasets.
>
> ***The real robot experiments are limited and do not report success rates.***
>
> Thank you for pointing this out. We have added quantitative success statistics for both real-robot tasks in the revised manuscript:
>
> - **ALOHA box flip:** 78% success rate (39/50). A trial is counted as successful if the robot flips the box to the target orientation within 500 control steps at 50 Hz.
> - **Franka fruit delivery:** 92% success rate (46/50) using the CLIP prompt reported in Appendix C.7. A trial is counted as successful if the robot correctly identifies the target fruit and anticipates and adapts to the human hand motion.
>
> These two real-robot experiments probe complementary challenges—non-smooth, unknown contact dynamics with external disturbances (ALOHA box flip) and adaptive human–robot interaction from a single demonstration with vision–language grounding (Franka fruit delivery). We clarify this in the text and provide an accompanying video for qualitative visualization.
>
> ***The paper does not compare to VINN (Pari et al. 2022), though cited as the closest prior work.***
>
> Thank you for this comment. While VINN is the closest prior work, GPI has a broader scope: VINN focuses on self-supervised visual representation learning combined with kNN-based action retrieval, whereas GPI combines flexible visual representations with an explicit geometric formulation for policy retrieval. At the representation level, our use of a VAE for latent embeddings is conceptually similar to VINN’s use of BYOL in that both are self-supervised encoders.
>
> In the revised manuscript, we therefore include a GPI+BYOL variant in Table 3, where we replace the VAE encoder with a BYOL encoder while keeping the GPI policy unchanged. In this setting, GPI+BYOL achieves an average score of 67%, compared to 88% with the VAE encoder. We believe this difference arises because BYOL emphasizes invariance to augmentations, whereas the VAE is trained to reconstruct inputs and thus yields a smooth latent manifold that better preserves continuous scene geometry (e.g., block and goal positions), which is particularly well suited for GPI’s distance fields and flows. This should not be interpreted as a general verdict on BYOL versus VAE, but rather as evidence that GPI can flexibly accommodate different self-supervised encoders. We have explicitly mentioned this connection in the revised manuscript.
>
> ***The paper does not show performances when demonstrations are suboptimal or contain mistakes.***
>
> In this work, we follow the standard setup of imitation-learning benchmarks, using the same demonstration datasets as prior work for fair comparison. We therefore do not explicitly construct additional “noisy” or heavily suboptimal demonstration sets. However, the demonstrations we use are not optimal: they are either human-provided or generated in simulation and can contain imperfections. The same holds for our real-world experiments, where demonstrations are manually collected and thus naturally non-optimal.
>
> Although we do not explicitly sweep over different levels of demonstration quality, GPI’s non-parametric, geometric structure is well-suited for such studies: the policy is built directly from individual demonstrations, and one can transparently inspect or modify which trajectories contribute to the resulting flow field. We view a systematic analysis of robustness to suboptimal or erroneous demonstrations, leveraging this structure, as a natural extension of our framework and an interesting direction for future work beyond the scope of the current work.

---

### Official Review · Reviewer_g86J · 2025-11-01

**Soundness:** 4
**Presentation:** 3
**Contribution:** 3
**Rating:** 6
**Confidence:** 4

**Summary:**

The paper proposes Geometry-aware Policy Imitation (GPI), which treats each demonstration as a geometric curve and builds a distance field in state space. Two complementary control primitives—(progression along the curve tangents and attraction via the negative distance gradient in the actuated subspace)—are superposed to form a lightweight, non-parametric vector-field policy. Multiple demonstrations are composed by distance-weighted retrieval (top-K with softmax temperature). This decouples metric learning (state representation / distance) from behavior synthesis (flow composition). Experiments on PushT (state + vision), RoboMimic (Lift/Can/Square) and Adroit (Door/Pen/Hammer/Relocate) plus two real-robot tasks (box flip on ALOHA, fruit handover on Franka) show GPI attains equal or higher success than diffusion policies while being 20–100× faster and more memory-efficient; a Lyapunov-style analysis argues convergence of the flow policy.

**Strengths:**

1.Clear geometric formulation & interpretability. Turning demonstrations into distance/flow fields yields an intuitive, controllable policy; the roles of attraction vs. progression and the top-K composition are explicit.

2.Efficiency. State-based PushT runs at ~0.6 ms per step vs. ~65 ms for DDIM-10; vision-based uses a light encoder with small memory. No training is needed for state-based settings.

3.Metric learning is flexible: task-specific heads, VAE latents, or pretrained features (ResNet/PCA, CLIP, SAM), and only robot-space distance shapes attraction—vision is used mainly for demo selection.

4.Theoretical sanity check. A concise Lyapunov argument establishes asymptotic convergence in the actuated subspace under the proposed dynamics

**Weaknesses:**

1.Metric dependence & high-dimensional brittleness. Performance hinges on the chosen distance; SAM underperforms without fine-tuning, and the method relies on learned embeddings for selection while only drob shapes attraction—raising questions under perceptual noise or representation drift.

2.Comparisons and settings. Diffusion baselines are strong but specific (DDPM-100/DDIM-10). It would help to compare against recent flow-matching/streaming-flow policy variants and to ensure action-space/horizon choices align fairly (the method uses H=1 reactive control).

3.Assumption clarity. The convergence proof assumes smooth distance fields and projection to the nearest point on a continuous curve (e.g., splines). Practicalities with discrete demos, non-holonomic constraints, and contact discontinuities are not fully analyzed.

**Questions:**

1.Retrieval & scaling. What is the exact complexity of one control step (in terms of demos N, trajectory lengths, and latent dimension)? Do you use approximate NN (e.g., FAISS) or down-sampling; how does latency scale to ~10⁶ states?

2.Adaptive weighting. You set λ₁, λ₂ constant or distance-dependent. Can these be scheduled online (e.g., larger attraction far from demos, or curvature-aware progression) and do you have guidelines for selecting β, K robustly across tasks?

3.Vision robustness. Since denv does not shape attraction, how do you mitigate mis-selection due to perceptual drift (e.g., lighting/occlusion)? Would a small attraction term in latent space (or a learned cross-term) help?

4.Baselines. Could you add comparisons to flow-matching policies and recent “streaming flow” formulations that also interpret action trajectories as flows? This would clarify whether speed/robustness advantages hold beyond diffusion.

---

> ### Author Response · Authors · 2025-11-21
> **Response to Reviewer g86J**
>
> Dear Reviewer g86J,
>
> Thank you for the valuable comments and suggestions. Our responses to your comments on convergence analysis, distance metrics, additional baselines, and scalability are provided in **General Response to All Reviewers (R1–R3)**. Below, we address your remaining concerns.
>
> ***Adaptive weighting. You set λ₁, λ₂ constant or distance-dependent. Can these be scheduled online (e.g., larger attraction far from demos, or curvature-aware progression) and do you have guidelines for selecting β, K robustly across tasks?***
>
> As noted below Eq. (2), $\lambda_1$ and $\lambda_2$ can be either constants or distance-dependent weights. In our PushT experiments, both choices give similar performance, suggesting that GPI is not overly sensitive to the precise schedule.
>
> The parameters $\beta$ and $K$ control how demonstrations are composed. Figure 5 compares different $K$ and shows that performance is stable across this range. Intuitively, as $\beta \to 0$, the policy smoothly blends the flow fields of the top$K$ demonstrations; as $\beta \to \infty$, it effectively follows the single closest demonstration. The new ablation over $\beta$ (Table 7) shows that reward remains high for $\beta$ between $10^{-4}$ and $10^{-1}$, further indicating robustness. On RoboMimic and Adroit, we simply fix $\beta = 10^{-4}$, $\lambda_1 = \lambda_2 = 1.0$, and $K = 3$ without additional tuning, which already achieves strong performance. As you suggest, these parameters could also be scheduled online to further improve performance, which we see as a natural extension.
>
> ***Vision robustness. Since $d_{\text{env}}$ does not shape attraction, how do you mitigate mis-selection due to perceptual drift (e.g., lighting/occlusion)? Would a small attraction term in latent space (or a learned cross-term) help?***
>
> We do not currently introduce an explicit mechanism that targets perceptual drift, but in the experiments we tested we found that GPI remains robust in practice. We believe two aspects of the design help:
>
> 1. **Reactive control.** GPI is fully reactive: at each control step, it re-retrieves the closest demonstrations and recomputes the flow. This continual re-selection lets the policy quickly correct small retrieval errors instead of committing to a long open-loop sequence.
>
> 2. **Attraction in robot space.** Perceptual drift only affects which demonstrations are selected and how they are weighted via $d_{\text{env}}$; it does not directly perturb the attraction term, which is defined in the robot/actuated subspace via $d_{\text{rob}}$. Thus, even if retrieval is slightly off due to visual noise, the flow still imitates some valid demonstration in the dataset, which tends to be stable and safe. In contrast, large parametric policy networks can sometimes produce qualitatively “off-manifold” behaviors under out-of-distribution inputs.
>
> Adding a small attraction term in latent space (or a learned cross-term) is indeed an interesting extension. Our current work does not yet include such a term, but it can be incorporated naturally by augmenting the attraction flow with a component defined in the visual feature space. Together with an appropriate latent dynamics model, this could further improve robustness to perceptual drift and enable richer stability and convergence analysis for visuomotor policies, which we view as promising future work.
>
> ***Baselines. Could you add comparisons to flow-matching policies and recent “streaming flow” formulations that also interpret action trajectories as flows? This would clarify whether speed/robustness advantages hold beyond diffusion.***
>
> The additional baselines are reported in “General Response to All Reviewers” (R3). Here, we briefly clarify how GPI differs from streaming flow policy (SFP): While both SFP and GPI interpret actions as flows, they obtain the flow in different ways. SFP uses a learned parametric model (trained in a flow-matching fashion) to predict a flow field over actions, whereas GPI is non-parametric and explicitly geometric. As a result, SFP inherits the usual training and capacity considerations of neural policy networks, while GPI retains the simplicity and interpretability of an explicit geometric construction, yet still achieves competitive or better performance and efficiency in our experiments.
>
> ***SAM underperforms without fine-tuning***
>
> We agree that SAM without fine-tuning does not achieve the best performance. However, we believe this is reasonable: its performance is likely limited by sensitivity to segmentation quality and by the downstream pose-estimation module. Even in that case, it achieves a 41% average score, indicating that GPI’s non-parametric policy synthesis can still leverage fairly generic segmentation features. In practice, SAM can be fine-tuned or adapted to the task to provide stronger latent features, as we do for other visual encoders. Therefore, we view our reported result as a conservative baseline rather than a limitation.

---

### Official Review · Reviewer_zwd9 · 2025-11-01

**Soundness:** 3
**Presentation:** 4
**Contribution:** 3
**Rating:** 6
**Confidence:** 3

**Summary:**

The paper proposes Geometry-Aware Policy Imitation, a method that treats expert demonstrations as geometric curves in state space instead of discrete state–action samples. Each demonstration defines a distance field, from which two flows are derived, a progression flow using demonstrated tangents, and an attraction flow using the negative gradient of the distance field. Their weighted sum forms a vector field that produces control commands. The paper states that this approach separates metric learning (state representation and distance computation) from behavior synthesis (flow combination). It presents a Lyapunov-based argument that the combined flow is stable in the actuated subspace and converges toward the demonstration. Experiments are reported on Push-T, Robomimic, Adroit and two real-robot tasks.

**Strengths:**

- Clear and consistent mathematical formulation.
- Demonstrated efficiency on Push-T.
- Comprehensive documentation of experimental settings, including both state-based and vision-based inputs.
- Modular structure allowing different encoders.
- The text includes explicit ablations on the number of neighbors, action horizon, and noise levels.

**Weaknesses:**

- The convergence proof covers only smooth, continuous trajectories in the actuated subspace. The paper does not provide analysis for environments with unactuated dynamics or discontinuities.
- The paper defines multiple possible distance metrics, but no quantitative comparison or sensitivity analysis across metrics is reported.
- Quantitative comparisons are limited to diffusion-based baselines.
- The paper gives descriptive outcomes and system specifications but no numerical success or robustness statistics in the real-robots experiments.
- The method requires storing features for all demonstrations. The text notes linear memory growth but provides no empirical scaling or retrieval-time measurements.
- Multimodality is illustrated through noise perturbations but not quantified beyond the presented plots.

**Questions:**

- Can the authors provide quantitative results showing how different distance metrics (Euclidean, geodesic, latent cosine) affect performance?
- Are there measured success rates or task completions for the ALOHA and Franka experiments?
- How does retrieval latency or memory use change with increasing numbers of demonstrations?
- How sensitive is the method to discontinuities or non-smooth demonstrations?
- Can the authors clarify whether diffusion-policy baselines were reimplemented under identical conditions or reused from prior work?

---

> ### Author Response · Authors · 2025-11-21
> **Response to Reviewer zwd9**
>
> Dear Reviewer zwd9,
>
> Thank you for your constructive feedback and for highlighting both the strengths and limitations of our work. Our responses to your comments on convergence analysis, distance metrics, additional baselines, and scalability are provided in **General Response to All Reviewers (R1–R3)**. Below, we address your remaining concerns.
>
> ***The paper gives descriptive outcomes and system specifications but no numerical success or robustness statistics in the real-robots experiments. Are there measured success rates or task completions for the ALOHA and Franka experiments?***
>
> Thank you for pointing this out. We have added quantitative success statistics for both real-robot tasks in the revised manuscript:
>
> - **ALOHA box flip:** 78% success rate (39/50). A trial is counted as successful if the robot flips the box to the target orientation within 500 control steps at 50 Hz. During these experiments, we also introduce occlusions and external disturbances; GPI still reliably completes the flip, indicating robustness to sensing and dynamics perturbations.
>
> - **Franka fruit delivery:** 92% success rate (46/50) using the CLIP prompt reported in Appendix C.7. A trial is counted as successful if the robot correctly identifies the target fruit and anticipates and adapts to the human hand motion. We vary the object shapes and initial positions while using only a single demonstration, and the policy consistently completes the task, demonstrating robustness and generalization.
>
> Please see the attached video for qualitative visualizations of both tasks.
>
> ***Multimodality is illustrated through noise perturbations but not quantified beyond the presented plots.***
>
> We would like to clarify that multimodality is not only illustrated qualitatively. It is also evaluated quantitatively in Fig. 6, where we report both the average reward and a trajectory-diversity metric as functions of the sampled noise level. As the noise increases, the induced policy covers more distinct behaviors rather than collapsing to a single mode, leading to higher trajectory diversity accompanied by a gradual reduction in reward. This trade-off curve provides a quantitative characterization of how GPI balances performance and multimodality as the stochasticity level is varied.
>
> ***Can the authors clarify whether diffusion-policy baselines were reimplemented under identical conditions or reused from prior work?***
>
> We confirm that all baselines were run using their official open-source implementations under identical evaluation conditions, unless otherwise specified in the paper. For example, for Diffusion Policy (DP) and Flow Matching Policy (FMP), we follow their original setup with an action horizon of $H = 8$, whereas Streaming Flow Policy (SFP) and GPI are run with $H = 1$.
>
> ---
> Finally, we hope that the added metric and weighting ablations, scaling analysis, and real-robot success statistics address your concerns about robustness and practicality, and we thank you again for helping us strengthen the paper.

---

### Author Response · Authors · 2025-11-21
**[1/3] General Response to All Reviewers**

We thank all reviewers for their thoughtful and constructive feedback. We are encouraged that they find our geometric formulation well-motivated, conceptually simple, and empirically effective. Below, we first address three recurring points and then respond to reviewer-specific comments.

### R1. Stability analysis for unknown / unactuated dynamics and non-smooth demonstrations

Our stability analysis in Appendix A is carried out under the hypotheses that the demonstrations define smooth curves in the actuated space and the induced distance field is differentiable. Under these assumptions, we prove asymptotic convergence in the actuated space to the demonstration manifold. The proof **does not apply** to discontinuous or time-discretized trajectories.

However, in the imitation learning benchmarks we evaluate (PushT, RoboMimic, Adroit), the demonstrations are time-discretized, so the smoothness assumptions of the Lyapunov theorem are not satisfied. In these settings, we therefore do not claim formal stability guarantees and only report empirical behavior, and the Lyapunov analysis in Appendix A should be interpreted as a supporting result for an idealized continuous setting rather than the main contribution. Despite the gap between the theoretical setting and the benchmarks, GPI still achieves high performance, efficiency, and exhibits qualitatively stable rollouts on all tasks we consider, suggesting that the geometric policy construction is robust in practice.

Regarding unactuated and unknown dynamics, our setting follows the standard imitation learning paradigm: environment dynamics and unactuated components (e.g., contact interactions, object motion, human motion) are treated as unknown, and policies are **learned purely from data rather than from a full dynamics model**. In this setting, obtaining rigorous convergence guarantees for the entire closed-loop system would require **additional assumptions** on scene dynamics that go beyond the scope of this work and are equally **unavailable** to the baselines we compare against. What GPI does provide, in contrast to these baselines, is an explicit geometric policy structure that is amenable to stability analysis in the actuated subspace. Thus, while full closed-loop guarantees under unknown environment dynamics remain an open challenge for the field, GPI contributes a principled foundation on top of which such analyses can be developed, and we view this analyzable structure as a key advantage of our formulation.

We have revised Appendix A and the Limitations section to state these assumptions explicitly.

---

> ### Author Response · Authors · 2025-11-21
> **[2/3] General Response to All Reviewers**
>
> ### R2. Distance metrics
>
> In all experiments, the metric used by GPI is constructed from simple and physically motivated components: Euclidean distance for robot states (e.g., joint angles, end-effector positions), geodesic distance for quaternions, and cosine similarity for latent visual embeddings. These choices follow the geometry and physical meaning of each state component. All state dimensions are normalized before distance computation, which further reduces sensitivity to manual tuning.
>
> To assess sensitivity to the specific form of the metric, we perform an ablation on the PushT task comparing L1 distance, L2 distance, and cosine similarity, both in the original state space and in a VAE-based latent space. The average rewards are summarized below (Table 5 in the revised paper):
>
> |       Metric      | State Space | Latent Space (VAE) |
> | :---------------: | :---------: | :----------------: |
> |  L1 norm distance |     88      |         81         |
> |  L2 norm distance |     86      |         85         |
> | Cosine similarity |     88      |         87         |
>
> In the state space, both L1 and L2 distances perform well, indicating that GPI is not sensitive to the exact choice of norm as long as the metric is consistent with the underlying state geometry. In the latent space, cosine similarity performs best, with L1 and L2 still competitive, consistent with standard practice for feature embeddings. Even when directly using cosine similarity in the state space, performance remains reasonable. Overall, these results indicate that GPI does not rely on fragile, hand-tuned metrics and works reliably with simple, task-aligned distances.
>
> We also investigated how the relative weighting between the environment distance $d_{\text{env}}$ and the robot distance $d_{\text{rob}}$ influences performance on PushT and RoboMimic tasks. We define the combined metric
>
> $d = w_{\text{rob}}\ d_{\text{rob}} + w_{\text{env}}\ d_{\text{env}}$
>
> and vary the ratio $w_{\text{env}}/w_{\text{rob}}$. The results (Table 6 in the revised paper) are:
>
> | $w_{\text{env}}/w_{\text{rob}}$ | 0.01 | 0.1  | 0.5  | 1.0  | 5.0  | 10.0 | 100.0 |
> | -------------------------------- | ---- | ---- | ---- | ---- | ---- | ---- | ----- |
> | PushT                             | 0.38 | 0.76 | 0.83 | 0.87 | 0.80 | 0.80 | 0.54  |
> | Lift                              | 0.68 | 0.85 | 0.98 | 1.00 | 0.83 | 0.78 | 0.52  |
> | Can                               | 0.61 | 0.73 | 0.88 | 0.96 | 0.79 | 0.58 | 0.18  |
> | Square                            | 0.23 | 0.55 | 0.72 | 0.82 | 0.63 | 0.51 | 0.16  |
>
> Performance is clearly degraded when the environment weight is extremely small or large (e.g., $w_{\text{env}}/w_{\text{rob}} = 0.01$ or $100$), but remains high over a broad intermediate range (approximately $w_{\text{env}}/w_{\text{rob}} \in [0.1, 10]$). These trends confirm that the metric and its weighting are important design choices, but also show that GPI is robust to a wide range of relative weightings between robot and environment distances.

---

> ### Author Response · Authors · 2025-11-21
> **[3/3] General Response to All Reviewers**
>
> ### R3. Baselines and scalability of GPI
> **Additional baselines**: We have added two recent and competitive generative policy baselines on the pushT task: Flow Matching Policy (FMP) [1] and Streaming Flow Policy (SFP) [2]. For all baselines (DDPM, DDIM, FMP, SFP), we use the same architecture as in our original diffusion-policy experiments, following their official public implementations. Detailed settings for these baselines are provided in the revised manuscript.
>
> | Method         | (Avg./Max.) score (State) | Training / Inference Time (State) | Memory (State) | (Avg./Max.) score (Vision) | Training / Inference Time (Vision) | Memory (Vision) |
> | -------------- | ------------------------: | ---------------------------------:| --------------:| --------------------------:| ----------------------------------:| ---------------:|
> | DDPM           |               82.3 / 86.3 |                    1.0 h / 641 ms |         252 MB |                80.9 / 85.5 |                     2.5 h / 647 ms |          353 MB |
> | DDIM           |               81.5 / 85.1 |                     1.0 h / 65 ms |         252 MB |                79.1 / 83.1 |                      2.5 h / 67 ms |          353 MB |
> | FMP            |               77.6 / 80.2 |                     1.0 h / 58 ms |         251 MB |                75.1 / 79.3 |                      2.5 h / 60 ms |          352 MB |
> | SFP            |               83.1 / 87.8 |                     0.8 h / 51 ms |         240 MB |                77.5 / 81.2 |                      2.0 h / 55 ms |          341 MB |
> | **GPI (Ours)** |           **85.8 / 89.0** |                  **0 h / 0.6 ms** |     **0.7 MB** |            **83.3 / 86.9** |                 **0.3 h / 3.3 ms** |       **44 MB** |
>
> As discussed in the revised paper, GPI achieves the best overall performance among all methods. Although FMP and SFP are more efficient than standard diffusion policies in terms of inference latency, they still rely on a parametric model to represent a policy or flow field, following an overall pipeline similar to diffusion-based approaches. In contrast, GPI fundamentally differs in its geometric formulation: the policy is an explicit construction over a task-relevant metric space rather than a learned action-generation network. This structural difference directly yields large gains in training cost (no policy-network training for state-based settings, and only a lightweight encoder for vision), inference speed, and memory efficiency.
>
> **Complexity of one control step and scalability**: All demonstrations are stored in a tensor of shape $(N T, D)$, where $N$ is the number of demonstrations, $T$ is the trajectory length, and $D$ is the state/feature dimension. Given the current observation of shape $(1, D)$, we compute its distance to all stored states in a single batched operation. The complexity of the retrieval step is therefore $O(N T D)$, implemented in parallel on the GPU.
>
> To make the scaling explicit, we report retrieval latency and memory usage as we vary the number of stored states from $10^2$ to $10^6$ and the feature dimension $D$ from 5 to 512. We measure the cost of computing distances between a single query state and all stored features, which is the component that grows with the database size. The results (Table 4 in the revised paper) are:
>
> | \# states        | $D = 5$ (ms / MB) | $D = 32$ (ms / MB) | $D = 128$ (ms / MB) | $D = 512$ (ms / MB) |
> |------------------|-------------------|--------------------|---------------------|----------------------|
> | $1\times 10^{2}$ | 0.039 / 0.00      | 0.033 / 0.01       | 0.034 / 0.05        | 0.033 / 0.20         |
> | $1\times 10^{3}$ | 0.039 / 0.02      | 0.036 / 0.12       | 0.035 / 0.49        | 0.039 / 1.96         |
> | $1\times 10^{4}$ | 0.033 / 0.19      | 0.037 / 1.22       | 0.068 / 4.88        | 0.217 / 19.6         |
> | $1\times 10^{5}$ | 0.051 / 1.91      | 0.221 / 12.2       | 0.517 / 48.8        | 1.973 / 196          |
> | $1\times 10^{6}$ | 0.291 / 19.1      | 1.920 / 122        | 4.891 / 488         | 19.26 / 1955         |
>
>
> Each entry reports latency (ms) / memory (MB). Even with very large databases ($10^6$ states) and high-dimensional features ($D = 512$), retrieval remains below 20ms with about 2GB of memory, which is compatible with typical real-time manipulation settings.
>
> Reference:
>
> [1] Fan Zhang and Michael Gienger. Affordance-based robot manipulation with flow matching. arXiv
> preprint arXiv:2409.01083, 2024.
>
> [2] Sunshine Jiang, Xiaolin Fang, Nicholas Roy, Tomas Lozano-Perez, Leslie Pack Kaelbling, and Siddharth Ancha. Streaming flow policy: Simplifying diffusion / flow-matching policies by treating
> action trajectories as flow trajectories. arXiv preprint arXiv:2505.21851, 2025.

---

### Author Response · Authors · 2025-12-01
**Summary of Revisions and Rebuttal**

Dear AC,

We briefly summarize how our rebuttal and the revised manuscript address the reviewers’ concerns:

# 1. Scope of the stability analysis
We clarified that the Lyapunov-style stability proof is formulated for an idealized continuous setting with smooth demonstrations and differentiable distance fields, and **does not claim** guarantees for the time-discretized trajectories used in our experiments. Nonetheless, GPI achieves **strong empirical performance** on all imitation-learning benchmarks we consider (PushT, RoboMimic, Adroit), where these assumptions do not formally hold, indicating robustness in practice. We also clarify that all compared methods operate in the same standard imitation-learning setting, learning purely from data without an explicit dynamics model. In such settings, closed-loop stability guarantees are typically difficult to obtain. The explicit geometric structure of our approach nevertheless enables a Lyapunov-style analysis in the actuated subspace under suitable assumptions, which we view as a first step toward more complete guarantees.

# 2. Distance metrics
We added new ablations that vary both the metric type (L1, L2, cosine, state vs. latent space) and the robot–environment weighting. These results show that performance remains high over a broad range of choices and only degrades for extreme weight ratios, indicating robustness to metric design rather than dependence on fragile hand-tuning.

# 3. Additional baselines
We incorporated two recent generative-policy baselines—Flow Matching Policy and Streaming Flow Policy—for a more comprehensive comparison under matched settings. In addition, following Reviewer aXzo’s suggestion, we added a variant that uses VINN-style BYOL features as the latent representation for the vision-based policy, clarifying how GPI behaves with that encoder.

# 4. Scalability
We now report retrieval latency and memory usage as the number of stored states increases up to $10^6$ and the feature dimension up to 512. The results show that GPI maintains low latency and memory requirements in this regime, demonstrating that the approach scales to large demonstration datasets.

# 5. Real-robot success rates
We now report quantitative success statistics for both real-robot experiments: 78% (39/50) success for the ALOHA box-flip task and 92% (46/50) success for the Franka fruit-delivery task, with clearly specified success criteria.

# 6. Other reviewer questions
We also addressed the remaining points raised by the reviewers, including multimodality quantification, vision robustness, and parameter choices, and clarified these aspects in the revised text.

We hope this summary helps you assess how the paper has improved after the rebuttal and revisions, and we sincerely appreciate your time and effort.

Best,

The Authors

---

### Meta-Review · Area_Chair_rphw · 2026-01-05

**Summary:**

This paper proposes to rethink imitation learning through geometry and introduces Geometry-Aware Policy Imitation, a method that treats expert demonstrations as geometric curves in state space instead of discrete state-action samples. Each demonstration defines a distance field, from which two flows are derived: (i) a progression flow using demonstrated tangents, and (ii) an attraction flow using the negative gradient of the distance field. These two flows are superposed to form a lightweight, non-parametric vector-field policy.
In terms of theoretical grounding, inspired by (Li and Calinon, 2025), this paper presents a Lyapunov-based analysis that the combined flow is stable in the actuated subspace and converges toward the demonstration. Extensive experiments on PushT, RoboMimic (including Lift, Can, and Square), Adroit (including Door, Pen, Hammer, and Relocate) and two real-robot tasks altogether show that GPI attains equal or higher success rates than diffusion policies while being much faster at inference and more memory-efficient.


The main strengths of this work:
- All reviewers appreciate the novelty of the geometric perspective on imitation learning, which is well-motivated and intriguing, and the proposed method is intuitive, reasonable, and theoretically grounded.
- Most of the reviewers appreciate the strong empirical performance of GPI, the extensive ablations and real-world validations of their approach.
- One notable strength is the policy inference speedup and memory reduction, which offer a significant advantage over the strong baselines like Diffusion Policy and Flow Matching Policy.
- The paper is overall clearly written, with a clear and consistent formulation and a helpful illustrative 2D example explaining the main idea.

At the same time, the reviewers also raised several concerns:

(1) Design of the distance metrics and the resulting sensitivity or robustness (Reviewers: zwd9, g86J, tvdn):

Multiple reviewers consistently emphasized that distance metric selection is a critical design choice, yet the paper provides limited quantitative sensitivity analysis nor validation across alternative metrics.

(2) Assumptions for the stability guarantees (Reviewers zwd9, g86J, aXzo, tvdn):

All the reviewers noted that the Lyapunov analysis assumes smooth distance fields, continuous trajectories, and ideal projections onto continuous curves, which may not hold in  practical robotic settings, such as discrete demonstrations, contact-rich manipulation, or environments with unactuated dynamics and discontinuities.

(3) More experimental evaluation and comparisons needed (Reviewers zwd9, g86J, aXzo):

(i) Several reviewers point out that quantitative comparisons are restricted primarily to diffusion-based baselines and that key related methods (e.g., VINN) are not evaluated despite being cited as close prior work. (ii) The reported success rates are often comparable to Diffusion Policy, and reviewers suggest using more challenging tasks to better expose performance differences.


(4) Real-world validation can be strengthened (Reviewers zwd9, aXzo, tvdn):

Reviewers also highlighted that (i) real-robot experiments report descriptive outcomes but omit numerical success rates, robustness statistics, or stress tests under strong perturbations such as collisions or sensor noises. (ii) The requirement to store features for all demonstrations raises scalability concerns. (iii) Robustness to suboptimal or noisy demonstrations is not explored.

**Reviewer Concerns:**

After taking a careful read through the rebuttal, I find all the concerns to be either fully addressed or largely alleviated.

Regarding (1), during the rebuttal, the authors added new ablations that involve both the choice of metric (L1, L2, cosine, state vs. latent space) and the robot–environment weighting. The additional results confirmed that performance remains largely unaffected over a variety of choices and only degrades for extreme cases.

As for (2), the authors clarify that the stability analysis for an idealized continuous setting serves mainly as theoretical grounding, and the paper does not claim guarantees for the time-discretized trajectories used in the experiments.

As for (3), in the rebuttal, two more recent baselines, namely Flow Matching Policy (FMP) and Streaming Flow Policy (SFP), are evaluated on the pushT task to strengthen the experiments. Regarding VINN, the authors clarified the difference in the scope of the algorithms (i.e., VINN for self-supervised visual representation learning vs GPI for policy retrieval with flexible visual representation) and instead include a new variant GPI+BYOL to support this.

Regarding (4), for the real-robot experiments, the authors reported quantitative success statistics for both the ALOHA box-flip task and the Franka fruit-delivery task. To address the scalability issue, the rebuttal also reported the retrieval latency and memory usage with the number of stored states.

**Reviewer Scores:**

In the initial reviews, the reviewers are all on the positive side (zwd9: 6 / g86J: 6 / aXzo: 6 / tvdn: 8)

Given that most of the aforementioned main concerns have been addressed by the rebuttal, I believe the reviewers would have still voted for acceptance unanimously if they had been in the discussion.

Overall this is a nice contribution to the imitation learning literature and can inspire more follow-up research.

---

### Decision · Program_Chairs · 2026-01-26

Accept (Poster)